# GO WITH YOUR GUT: SCALING CONFIDENCE FOR AUTOREGRESSIVE IMAGE GENERATION

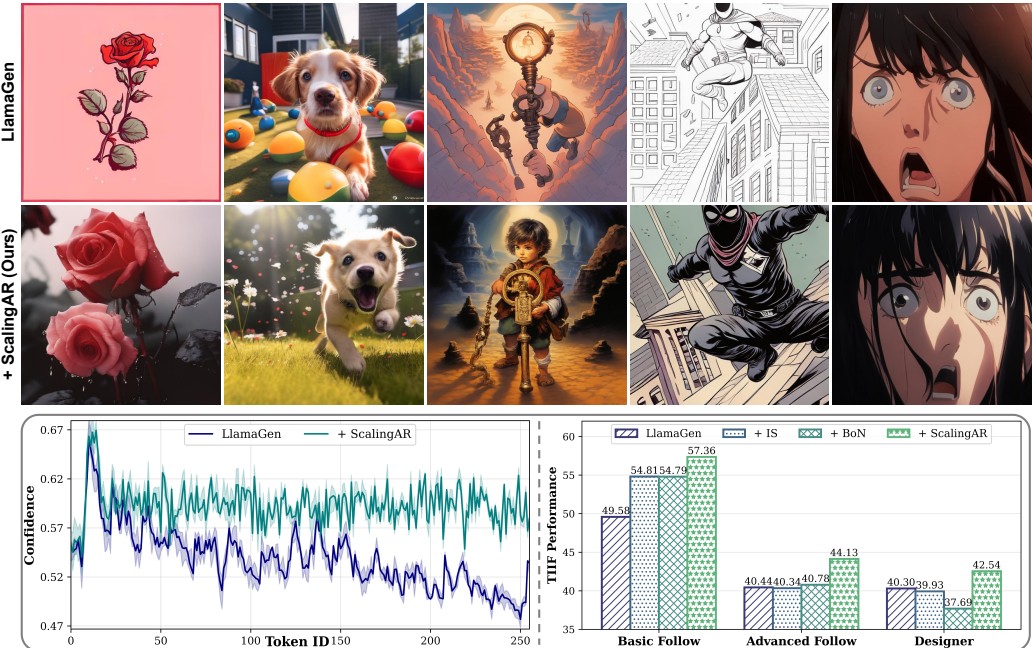

Figure 1: (***Top***) `ScalingAR` significantly improves the quality of autoregressive image generation. Detailed prompts are provided in **Appendix** §A. (***Bottom Left***) The token confidence trajectory over the generation process. (***Bottom Right***) Performance comparison of `ScalingAR` on TIIF-Bench with classic test-time scaling strategies, *i.e.*, Importance Sampling (IS) and Best-of-N (BoN).

## ABSTRACT

Test-time scaling (TTS) has demonstrated remarkable success in enhancing large language models, yet its application to next-token prediction (NTP) autoregressive (AR) image generation remains largely uncharted. Existing TTS approaches for visual AR (VAR), which rely on frequent partial decoding and external reward models, are ill-suited for NTP-based image generation due to the inherent incompleteness of intermediate decoding results. To bridge this gap, we introduce `ScalingAR`, the first TTS framework specifically designed for NTP-based AR image generation that eliminates the need for early decoding or auxiliary rewards. `ScalingAR` leverages *token entropy* as a novel signal in visual token generation and operates at two complementary scaling levels: (***i***) ***Profile Level***, which streams a calibrated confidence state by fusing intrinsic and conditional signals; and (***ii***) ***Policy Level***, which utilizes this state to adaptively terminate low-confidence trajectories and dynamically schedule guidance for phase-appropriate conditioning strength. Experiments on both general and compositional benchmarks show that `ScalingAR` (**1**) improves base models by $12.5\%$ on GenEval and $15.2\%$ on TIIF-Bench, (**2**) efficiently reduces visual token consumption by $62.0\%$ while outperforming baselines, and (**3**) successfully enhances robustness, mitigating performance drops by $26.0\%$ in challenging scenarios. Our code will be released in `ScalingAR Repository`.

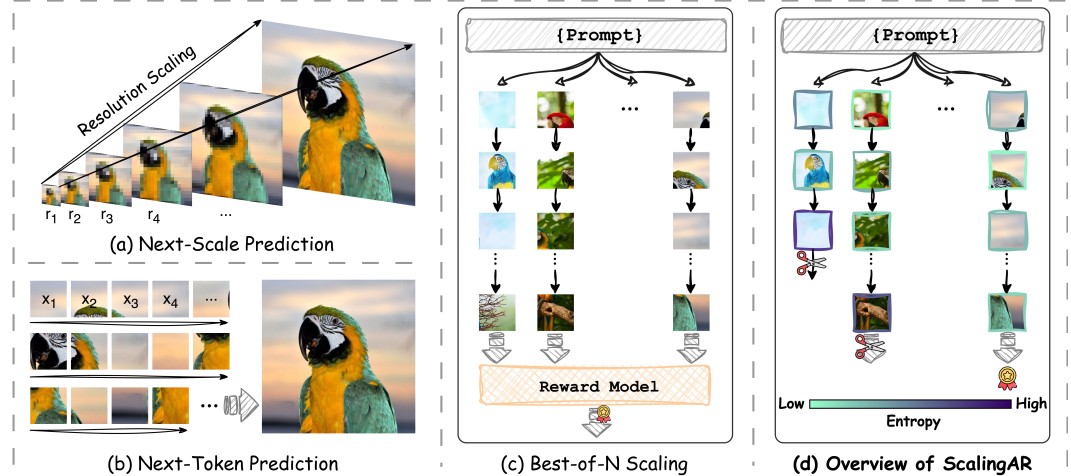

Figure 2: (**a**) Next-scale prediction paradigm generates multi-scale token maps coarse-to-fine. (**b**) Next-token prediction paradigm sequentially predicts next image tokens. (**c**) Illustration of Best-of-N sampling that generates multiple candidate and selects the best via voting or scoring. (**d**) Overview of our proposed `ScalingAR`, highlighting its ability to leverage token entropy to early-stop low-confidence samples and identify winning samples without the need for additional reward models.

# 1 INTRODUCTION

Large language models (LLMs) (Brown et al., 2020; Vaswani et al., 2017; Radford et al., 2019) have demonstrated the capabilities of next-token prediction (NTP) paradigm. This success has renewed interest in applying autoregressive (AR) architectures beyond text, motivating recent visual generative models that represent images in discrete token spaces (Sun et al., 2024; Tian et al., 2024; Li et al., 2024) as shown in Figure 2 (*b*). Compared to diffusion models, which operate over continuous noise trajectories, token-based AR models promise a more unified modality interface.

As the field evolves, the parameters and training data of foundation models (Wang et al., 2024; Yang et al., 2025) have increasingly grown to levels that are inaccessible for most university researchers. In this context, many studies have started to investigate **post-training** methods. Inspired by recent advancements such as GRPO (Shao et al., 2024), a surge of reinforcement learning research has emerged in both language and visual generation domains (Jiang et al., 2025; Cui et al., 2025). Meanwhile, another research avenue focusing on **test-time scaling** (TTS) has emerged (Lightman et al., 2023; Muennighoff et al., 2025; Zuo et al., 2025), which aims to explore *whether a slight increase in computational expense during inference can achieve performance on par with training-time methods, which typically incur much larger costs.*

While test-time scaling has been extensively researched in language models, analogous progress for autoregressive visual generation remains sparse. Images differ from text in three practical ways that complicate direct transfer: (*i*) **holism:** dropping the last 20% of a text sequence may still leave a syntactically valid answer, whereas truncating an image token stream yields an unusable artifact; (*ii*) **objective ambiguity:** many language scaling setups optimize toward a verifiable final answer (*e.g.*, math reasoning), whereas image generation lacks a single ground-truth target; and (*iii*) **early signal scarcity:** partial image token decodes are visually unstable, making premature selection risky. Moreover, recent work TTS-VAR (Chen et al., 2025b) introduced TTS for the next-scale prediction (NSP) paradigm in visual autoregressive model (VAR) (Tian et al., 2024) by predicting images in a coarse-to-fine manner (Figure 2 (*a*)). This intermediate visibility enables reward models to score during scaling but comes with limitations that require predicting large residual token maps at each scale and frequent decoding makes the process inefficient and less suitable for the NTP paradigm.

Building on these insights, we introduce `ScalingAR`, the first test-time scaling framework tailored to the NTP paradigm in autoregressive image generation. Unlike next-scale TTS-VAR, `ScalingAR` eliminates the need for frequent partial decoding and external reward models (as shown in Figure 2 (*d*)), relying solely on intrinsic signals derived from visual **token entropy** and conditional signals to profile confidence. Specifically, in response to limitations, `ScalingAR` prunes unreliable trajectories

without interrupting generation (*holism*), constructs confidence by combining intrinsic uncertainty and conditional signals (*objective ambiguity*), and extracts stability directly from model probabilities rather than intermediate outputs (*early signal scarcity*). Technically, `ScalingAR` features a two-level design: ❶ *Profile Level*, which constructs a unified confidence state by integrating intrinsic generation stability with conditioning effectiveness; and ❷ *Policy Level*, which leverages this confidence state to prune failing trajectories and dynamically adjust conditioning strength through adaptive termination and guidance scheduling. Our contributions can be summarized as follows:

- We propose `ScalingAR`, the first test-time scaling framework tailored to next-token prediction AR image generation, featuring a novel two-level design with Profile Level for dual-channel confidence profiling on-the-fly, and Policy Level for trajectory pruning and guidance scheduling.

- We for the first time investigate token entropy in visual token generation. By relying solely on intrinsic signals from the model, `ScalingAR` eliminates the need for frequent early decoding and external reward models, enabling a more efficient and reliable scaling process.

- Extensive experiments on both general and compositional benchmarks demonstrate that `ScalingAR` is: (*i*) **high-performing**, achieving significant performance gains over base models (*i.e.*, Llam-aGen and AR-GRPO), by $12.5\%$ on GenEval and $15.2\%$ on TIIF-Bench; (*ii*) **token-efficient**, outperforming classic baselines (*i.e.*, Importance Sampling and Best-of-N) while reducing visual token consumption by $62.0\%$; and (*iii*) **robust in challenging scenarios**, mitigating performance degradation by $26.0\%$ compared to base models in highly complex generation settings.

## 2 RELATED WORK

**Autoregressive Image Generation**   Autoregressive models have leveraged the scaling capabilities of language models (Yang et al., 2025; Brown et al., 2020; Radford et al., 2019) to generate images. These approaches employ discrete image tokenizers (Van Den Oord et al., 2017; Razavi et al., 2019) in conjunction with transformers, using a next-token prediction strategy. VQ-based methods (Lee et al., 2022; Razavi et al., 2019; Esser et al., 2021), *e.g.*, VQ-VAE (Van Den Oord et al., 2017), convert image patches into index-based tokens, which are then predicted sequentially by a decoder-only transformer. However, these VQ-based AR methods are limited by the lack of scaled-up transformers and the inherent quantization error in VQ-VAE. This has prevented them from achieving performance on par with diffusion models. Recent advancements (Wu et al., 2025a; Yu et al., 2022; Team, 2024) have scaled up AR models for visual generation. Additionally, some variants have been proposed, such as the next-scale prediction paradigm of VAR (Tian et al., 2024; Han et al., 2025), which predicts from coarse to fine token maps, and the parallel token prediction of masked AR (MAR) (Li et al., 2024; Wu et al., 2025b; Fan et al., 2025). Despite these developments, the mainstream approach remains the NTP paradigm, particularly as the field moves towards unified models (Xie et al., 2025; Wang et al., 2024; Ge et al., 2024) that can jointly handle textual and visual tokens. This alignment with language modeling allows for more versatile and scalable architectures.

**Test-Time Scaling**   Current LLMs have increasingly succeeded by allocating substantial reasoning at inference time, a paradigm known as test-time scaling (Snell et al., 2024; Welleck et al., 2024). This scaling can occur along two main axes: (1) Chain-of-Thought (CoT) (Wei et al., 2022) Depth: lengthening a single reasoning trajectory through more thinking steps, often relying on large-scale reinforcement learning with many samples (Yang et al., 2025; Jaech et al., 2024; Guo et al., 2025) or simpler post-training strategies (Ye et al., 2025; Muennighoff et al., 2025); (2) Parallel Generation: scaling by increasing the number of trajectories and aggregating them, as seen in works like Self-Consistency (Wang et al., 2023) and Best-of-N (Lightman et al., 2023). Recent efforts (Kang et al., 2025; Fu et al., 2025) have also integrated confidence estimation through token entropy into the test-time reasoning process, allowing the quality of individual traces to be assessed before aggregation with the rewards for majority voting (Wang et al., 2023). However, exploring TTS for AR image generation has been limited. This is due to the holistic nature of image generation, where overall coherence is paramount (see Figure 2 (*c*)), unlike reasoning tasks with well-defined ground truths. Additionally, the frequent early decoding required for images can be more computationally expensive than for language, suggesting that direct transfer of many LLM TTS techniques may not be suitable or optimal. To address this gap, we propose the first TTS strategy tailored for AR image generation. Notably, we pioneer the exploration of token entropy in image generation, enabling our method to leverage visual token confidence without the need for early decoding or additional rewards.

## 3 PRELIMINARIES

**Next-Token Prediction Autoregressive Modeling**    NTP is a fundamental paradigm in autoregressive models, where the model generates sequences by predicting the next token based on previously generated tokens. The generation process can be mathematically described as follows:

$$p(x_1, x_2, \ldots, x_T) = \prod_{t=1}^{T} p(x_t | x_1, x_2, \ldots, x_{t-1}). \tag{1}$$

This formulation allows the model to leverage past information to inform future predictions, making it particularly effective for sequential data generation.

The training of autoregressive models typically involves maximizing the likelihood of the observed sequences, which can be expressed as:

$$L = \sum_{t=1}^{T} \log p(x_t | x_{<t}). \tag{2}$$

This objective encourages the model to learn the underlying distribution of the data, enabling it to generate coherent and contextually appropriate sequences.

**Token Entropy in Language Modeling**    Token entropy is a critical metric for evaluating the uncertainty associated with the predictions made by language models (Kang et al., 2025). It quantifies the amount of unpredictability in the model's output distribution for a given token. The entropy $H$ at a specific position $i$ in the sequence can be defined as:

$$H_i = -\sum_{j} p_i(j) \log p_i(j), \tag{3}$$

where $p_i(j)$ denotes the predicted probability of the $j$-th token in the vocabulary at position $i$. Low entropy indicates high certainty in the prediction, while high entropy reflects greater uncertainty.

Furthermore, token confidence can be derived from the predicted distribution (Fu et al., 2025). The confidence $C_i$ for a token at position $i$ is defined as:

$$C_i = -\frac{1}{k} \sum_{j=1}^{k} \log p_i(j), \tag{4}$$

where $k$ represents the number of top tokens considered. High confidence values correlate with sharper distributions, indicating that the model is more certain about its predictions.

## 4 METHODOLOGY

To pioneer test-time scaling for next-token prediction autoregressive image generation, we propose `ScalingAR`, which leverages intrinsic token confidence signals without relying on early decoding or external rewards, featuring two scaling levels: (**i**) Dual-Channel Confidence Profile compacts heterogeneous per-step signals into a calibrated confidence state (§4.1); and (**ii**) Confidence-Guided Policies act on this state to prune failing trajectories and adapt conditional guidance on-the-fly (§4.2).

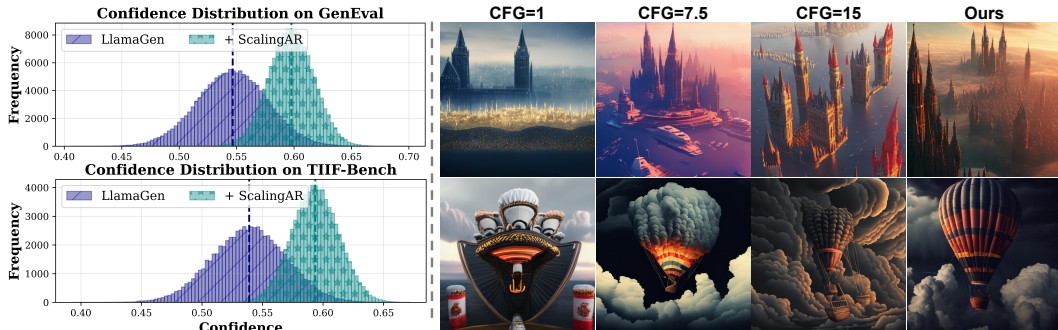

Figure 3: (**Left**) Confidence distribution of `ScalingAR` on GenEval and TIIF-Bench. (**Right**) Illustration of the trade-off between visual quality and semantic alignment with fixed Classifier-Free Guidance (CFG) in AR image generation. *1st*: *A 35 mm photo of a cityscape resembling Moscow floating in the sky on flying islands. 2nd*: *The colorful hot air balloon floated near the dark grey storm clouds.*

## 4.1 DUAL-CHANNEL CONFIDENCE PROFILE

Autoregressive image generators traditionally treat all partial trajectories as equally promising until completion, as illustrated in Figure 2 (*c*). However, empirical inspection reveals two dominant failure modes during inference that often foreshadow poor final results: ❶ **local intrinsic instability**, characterized by high entropy pockets and wavering token choices (Figure 1 (*Bottom Left*) & Figure 3 (*Left*)); and ❷ **poor utilization of the text condition**, where the semantic influence of the prompt gradually fades, resulting in misaligned or aesthetically suboptimal outputs (Figure 3 (*Right*)).

To address these challenges, we introduce the Dual-Channel Confidence Profile, consisting of two complementary channels: ① *Intrinsic Channel*: Captures localized instability and spatial anomalies within the token grid. ② *Conditional Channel*: Quantifies the marginal contribution of textual conditioning to ensure semantic alignment.

### 4.1.1 INTRINSIC CHANNEL: UNCERTAINTY & SPATIAL STABILITY

Early-stage failures in autoregressive image generation rarely manifest as immediate global collapse. Instead, they emerge through localized instability. To capture these signals, the Intrinsic Channel integrates two key components: token-level confidence and worst-block spatial stability.

**Token-level Confidence** Token-level uncertainty reflects the dispersion and decisiveness of predictions at each decoding step. Let $\pi_t$ denote the softmax distribution over the vocabulary $V$ at step $t$. We compute token entropy $H_t = -\sum_{v \in V} \pi_t(v) \log \pi_t(v)$ and top-1/top-2 margin $m_t = \pi_t(v_1) - \pi_t(v_2)$, forming a normalized uncertainty surrogate:

$$\widehat{H}_t = H_t / \log |V|, \quad u_t = \alpha_H \widehat{H}_t + \alpha_M(1 - m_t), \quad \alpha_H + \alpha_M = 1, \tag{5}$$

where $u_t$ is mapped to token confidence $s_t^{\text{tok}} = 1 - u_t \in (0, 1]$. To stabilize this signal, we apply an exponential moving average (EMA):

$$\bar{s}_t^{\text{tok}} = (1 - \lambda_{\text{tok}})\bar{s}_{t-1}^{\text{tok}} + \lambda_{\text{tok}} s_t^{\text{tok}}. \tag{6}$$

**Worst-block Stability** Localized "hot spots" of persistent high entropy often diffuse into global semantic corruption. To capture these spatial anomalies, we partition the $h \times w$ token grid into non-overlapping $b \times b$ blocks. For each block $k$ (with fill ratio $\geq \rho_{\min}$), we compute its mean normalized entropy $E_k$. Focusing on the worst-$q\%$ subset $W_t$ of blocks with the highest entropy:

$$E_{\text{worst}}(t) = \frac{1}{|W_t|} \sum_{k \in W_t} E_k. \tag{7}$$

A rolling min-max normalization $N_{\text{mm}}$ yields a stability score $B_t = 1 - N_{\text{mm}}(E_{\text{worst}}(t))$, emphasizing emergent localized failure rather than global averages.

Finally, the Intrinsic Channel score combines token-level confidence and worst-block stability:

$$I_t^{\text{raw}} = w_{\text{tok}}\bar{s}_t^{\text{tok}} + w_{\text{blk}}B_t, \quad w_{\text{tok}} + w_{\text{blk}} = 1, \tag{8}$$

followed by smoothing $I_t = \text{EMA}(I_t^{\text{raw}}, \lambda_I)$.

### 4.1.2 CONDITIONAL CHANNEL: TEXT UTILIZATION STRENGTH

While intrinsic signals capture localized instability, semantic misalignment often arises from insufficient utilization of the text condition. For concise prompts or complex visual contexts, the conditional branch may lose influence, silently drifting from the intended semantics. The Conditional Channel measures the marginal contribution of textual conditioning to ensure semantic alignment.

Let $p_{c,t}$ and $p_{u,t}$ denote the softmax distributions from conditional and unconditional logits, respectively. We compute the KL divergence $K_t = \text{KL}(p_{c,t} \parallel p_{u,t})$, then apply a rolling z-score normalization:

$$K_t^{\text{norm}} = \frac{K_t - \mu_K}{\sigma_K + \varepsilon}, \quad K_t^{\text{clip}} = \text{clip}(K_t^{\text{norm}}, -z_{\max}, z_{\max}), \tag{9}$$

mapping the result to $[0, 1]$:

$$\widehat{D}_t = 0.5 + 0.5 \frac{K_t^{\text{clip}}}{z_{\max}}. \tag{10}$$

Persistently low values of the smoothed score $\widehat{D}_t$ flag semantic fade, while excessively high values paired with low $I_t$ may indicate unstable over-conditioning.

### 4.1.3 UNIFIED CONFIDENCE STATE

To enable dynamic trajectory control, we combine both channels into a unified confidence state. The scalar unified confidence score is defined as:

$$C_t = w_I I_t + w_D \widehat{D}_t, \quad w_I + w_D = 1, \tag{11}$$

optionally passed through an affine-sigmoid calibration to mitigate cross-prompt scale drift. To capture early-stage failure signals, we maintain the running minimum $C_{\min}(t) = \min_{i \leq t} C_i$ and compute a relative rebound:

$$R_t = \frac{C_t - C_{\min}(t)}{|C_{\min}(t)| + \varepsilon}. \tag{12}$$

This unified confidence score serves as the basis for dynamic trajectory pruning and adaptive conditioning, enabling efficient test-time scaling tailored to the NTP paradigm.

### 4.2 CONFIDENCE-GUIDED POLICIES

With a calibrated confidence score $C_t$, we transition from passive observation to *active test-time control*, enabling dynamic intervention in autoregressive generation. To achieve this, we introduce two lightweight yet effective policies: ① an *Adaptive Termination Gate* that prunes unpromising trajectories to reclaim computation; and ② a *Guidance Scheduler* that dynamically modulates CFG scale to balance semantic alignment.

#### 4.2.1 ADAPTIVE TERMINATION GATE

Failing trajectories often exhibit prolonged spans of low confidence, lingering in a "confidence basin" before producing final tokens that posterior reranking would discard. The Adaptive Termination Gate proactively terminates such trajectories, reclaiming computational resources.

**Threshold Initialization and Adaptation**  To identify failing trajectories, we initialize a confidence threshold $\theta_\downarrow$ after a warm-up period of $W_0$ steps. The threshold is set to the **p**-quantile ($\mathbf{p} \in [0.15, 0.25]$) of the collected $C_t$ values across active trajectories. This ensures that pruning targets the bottom-performing trajectories without prematurely terminating promising ones. The threshold is periodically updated every $\Delta_{\text{upd}}$ steps using an EMA-based adaptation:

$$\theta_\downarrow \leftarrow (1 - \lambda_\theta)\theta_\downarrow + \lambda_\theta \text{Quantile}_{\mathbf{p}}(\{C_t\}_{\text{recent}}). \tag{13}$$

where $\{C_t\}_{\text{recent}}$ denotes the confidence scores from recent decoding steps.

**Recovery Safeguard**  To mitigate false positives caused by transient dips in $C_t$, we incorporate a recovery mechanism. A trajectory is permitted to recover if it satisfies either of the following conditions within a recovery window $\Delta_{\text{rec}}$: (*a*) $C_t \geq C_{\min}(t) + \delta_{\text{rec}}$: absolute confidence rebound exceeds a pre-defined gap. (*b*) $R_t \geq r_{\text{thr}}$: relative rebound exceeds a threshold, indicating stabilization. Only trajectories failing both criteria are marked for termination.

**Termination Rule**  Once the protection horizon $T_{\min}$ (*e.g.*, 10% of $T$) has elapsed, a trajectory is terminated if $C_{\min}(t) < \theta_\downarrow$ and no recovery within last $\Delta_{\text{rec}}$ steps. Additionally, a hard-fail guard ($C_t < C_{\text{hard}}$) triggers immediate termination for catastrophic collapse scenarios, ensuring robustness against extreme failures. By over-initializing $K_{\text{target}} + M_{\text{buf}}$ trajectories and relying on pruning, we refine the candidate set without spawning replacements.

#### 4.2.2 GUIDANCE SCHEDULER

Fixed CFG scales enforce a static trade-off between semantic alignment and diversity, yet the "optimal" balance varies across decoding phases. The Guidance Scheduler dynamically adjusts the CFG scale $s_t$ based on real-time signals from the unified confidence profile.

The scheduler integrates three key signals to adapt $s_t$:

- **Conditional Utilization ($\widehat{D}_t$):** Low $\widehat{D}_t$ flags under-conditioning, prompting an increase in $s_t$ to reinforce prompt influence.

- **Intrinsic Volatility ($\text{Var}_{\text{recent}}(I)$):** High short-term volatility in $I$ indicates instability, warranting temporary bolstering of conditioning.

- **Rebound ($R_t$):** Strong rebounds suggest stabilized semantics, allowing $s_t$ to ease pressure and preserve diversity.

Table 1: Evaluation on GenEval (Ghosh et al., 2023) and TIIF-Bench (Wei et al., 2025) benchmarks. "Diff.+AR" refers to the unified architecture, and "MAR" indicates the masked AR architecture (Li et al., 2024). We **bold** the best results, and "↑" denotes that higher is better.

| Method | #Params | Arch. | GenEval | | | | TIIF-Bench | | | |
|---|---|---|---|---|---|---|---|---|---|---|
| | | | Two Obj.↑ | Posit.↑ | Color Attr.↑ | Over.↑ | Basic↑ | Advanced↑ | Designer↑ | Over.↑ |
| DALLE·3 (Betker et al., 2023) | - | Diff. | - | - | - | 0.67 | 78.40 | 68.45 | 62.69 | 72.94 |
| Show-o (Xie et al., 2025) | 1.3B | Diff.+AR | 0.80 | 0.31 | 0.50 | 0.68 | 71.30 | 59.89 | 68.66 | 59.24 |
| LightGen (Wu et al., 2025b) | 0.8B | MAR | 0.65 | 0.22 | 0.43 | 0.62 | 53.99 | 45.76 | 59.70 | 46.42 |
| Infinity (Han et al., 2025) | 2B | VAR | 0.85 | 0.49 | 0.57 | 0.73 | 71.63 | 57.81 | 61.19 | 59.66 |
| Emu3 (Han et al., 2025) | 8.5B | AR | 0.81 | 0.49 | 0.45 | 0.66 | - | - | - | - |
| Janus (Wu et al., 2025a) | 1.5B | AR | 0.68 | 0.46 | 0.42 | 0.61 | - | - | - | - |
| AR-GRPO (Yuan et al., 2025) | 0.8B | AR | 0.27 | 0.02 | 0.03 | 0.31 | 19.59 | 14.91 | 17.91 | 16.22 |
| + IS | 0.8B | AR | 0.47 | 0.08 | 0.07 | 0.44 | 26.00 | 19.03 | 17.62 | 19.84 |
| + BoN | 0.8B | AR | 0.46 | 0.08 | 0.06 | 0.44 | 25.67 | 19.91 | 20.69 | 21.08 |
| + ScalingAR (Ours) | 0.8B | AR | **0.54** | **0.24** | **0.15** | **0.49** | **29.71** | **26.43** | **25.90** | **26.35** |
| LlamaGen (Sun et al., 2024) | 0.8B | AR | 0.34 | 0.21 | 0.04 | 0.32 | 49.58 | 40.44 | 40.30 | 40.35 |
| + IS | 0.8B | AR | 0.21 | 0.11 | 0.02 | 0.14 | 54.81 | 40.34 | 39.93 | 42.44 |
| + BoN | 0.8B | AR | 0.27 | 0.11 | 0.02 | 0.15 | 54.79 | 40.78 | 37.69 | 42.02 |
| + ScalingAR (Ours) | 0.8B | AR | **0.40** | **0.28** | **0.12** | **0.36** | **57.36** | **44.13** | **42.54** | **46.47** |

Using these signals, we compute the raw CFG scale adjustment:

$$s_t^{\text{raw}} = s_{\text{base}} + \alpha(1 - \widehat{D}_t) + \beta\text{Var}_{\text{recent}}(I) - \gamma R_t, \tag{14}$$

where $\alpha, \beta, \gamma$ control the relative influence of each term. The final scale $s_t$ is smoothed and clamped to prevent excessive fluctuations:

$$s_t = \text{clamp}((1 - \lambda_{\text{cfg}})s_{t-1} + \lambda_{\text{cfg}}s_t^{\text{raw}}, s_{\text{min}}, s_{\text{max}}), \tag{15}$$

with a deadband ($|s_t - s_{t-1}| < \epsilon_s$) suppressing jitter to ensure stability.

## 5 EXPERIMENTS

In this section, we conduct extensive experiments to answer the following research questions: (**RQ1**) Does ScalingAR enhance the quality of generated images? (**RQ2**) Does ScalingAR outperform other TTS strategies for both effectiveness and efficiency? (**RQ3**) How sensitive is ScalingAR to its key components? (**RQ4**) Whether ScalingAR holds advantages over other TTS strategies in terms of both scalability and robustness?

### 5.1 EXPERIMENTAL SETTINGS

**Baselines** We apply ScalingAR to the advanced models: LlamaGen ($512 \times 512$) (Sun et al., 2024) and AR-GRPO ($256 \times 256$) (Yuan et al., 2025). Since no prior work has explored TTS for the NTP image generation, we focus our comparisons on the following conventional baselines: Importance Sampling (IS) (Owen & Zhou, 2000) and Best-of-N (BoN) (Lightman et al., 2023). We also provide results from Show-o (Xie et al., 2025), LightGen (Wu et al., 2025b), Infinity (Han et al., 2025), Emu3 (Wang et al., 2024), Janus (Wu et al., 2025a), and DALLE·3 (Betker et al., 2023) for reference.

**Evaluations** To evaluate the effectiveness of ScalingAR, we adopt GenEval (Ghosh et al., 2023) and TIIF-Bench (Wei et al., 2025) as primary benchmarks for both general and compositional text-to-image generation capabilities. These benchmarks offer a comprehensive evaluation of the model's ability to produce high-quality and semantically consistent images from text prompts.

### 5.2 PERFORMANCE & EFFICIENCY COMPARISON

To answer **RQ1** and **RQ2**, we comprehensively compare ScalingAR against two baselines on general and compositional benchmarks in Table 1, alongside qualitative results, user study, and token consumption comparisons shown in Figure 1, 4, and Figure 5. Key observations are summarized as follows: **Obs.❶** *ScalingAR excels in enhancing both general and compositional generation quality.* As illustrated in Table 1, our ScalingAR consistently outperforms baseline methods (*i.e.*, IS and BoN), which achieve minimal or even negative performance gains, across benchmarks targeting distinct aspects of text-to-image generation. Figure 1 (*Top*) and Figure 4 provide qualitative evidence of ScalingAR's capabilities, showcasing visually superior results that excel in aesthetic quality and semantic alignment, *e.g.*, numerical accuracy, color fidelity, and subject clarity. Furthermore,

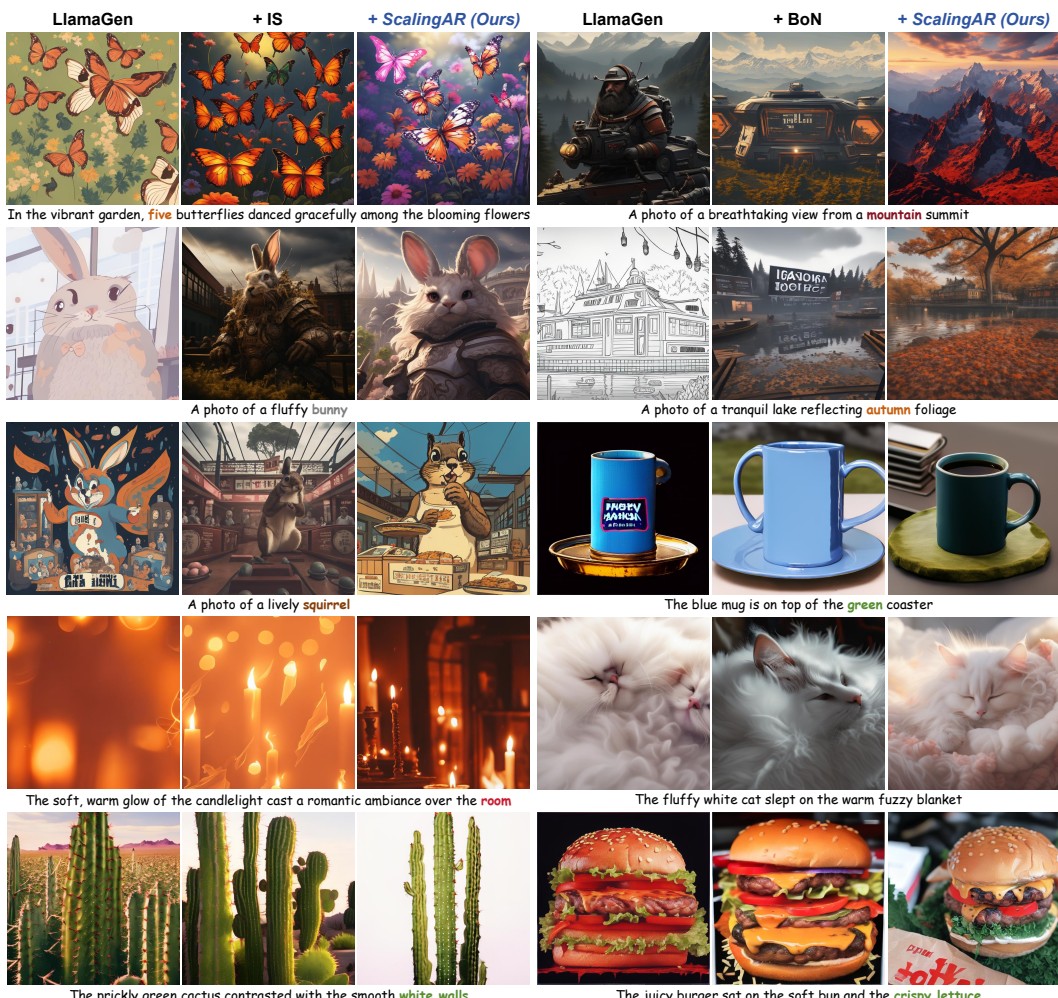

Figure 4: Qualitative results of `ScalingAR`. More results on AR-GRPO are provided in **Appendix** §E.

Figure 5 (*Left*) highlights `ScalingAR`'s effectiveness in aligning image generation with human preferences, as validated through user studies. **Obs.❷** *`ScalingAR` is a token-efficient test-time AR image generation enhancer.* Figure 5 (*Middle*) demonstrates that `ScalingAR` consistently surpasses other TTS strategies across benchmarks, requiring fewer visual tokens. Unlike BoN, which relies on external reward models and excessive token consumption, `ScalingAR` leverages intrinsic confidence signals to reduce computational overhead while maintaining high-quality outputs.

### 5.3 ABLATION ANALYSIS

To answer **RQ3**, we perform *step by step* evaluations on TIIF-Bench to analyze the contributions of `ScalingAR`'s confidence profiles, as detailed in Table 2. We give the following observations: **Obs.❸** *Effectiveness of Intrinsic Signal Profiling.* Removing Token-Level Confidence or Worst-Block Stability

Table 2: Ablation study of `ScalingAR`.

| Method | Bas.↑ | Adv.↑ | Des.↑ | Over.↑ |
|---|---|---|---|---|
| `ScalingAR` | **57.4** | **44.1** | **42.5** | **46.5** |
| w/o Conditional Channel | 54.1 | 43.1 | 42.2 | 45.1 |
| w/o Worst-Block Stability | 52.3 | 41.8 | 41.4 | 44.0 |
| w/o Token-Level Confidence | 49.6 | 40.4 | 40.3 | 40.4 |

both lead to a noticeable drop in performance, highlighting their critical role in capturing fine-grained entropy signals during visual token generation. This demonstrates the effectiveness of intrinsic signal profiling for maintaining local token stability and ensuring high-quality generation. **Obs.❹** *Importance of Condition State Balance.* Table 2 also reveals that removing the Conditional Channel leads to significant degradation. Figure 3 (*Right*) further confirms its critical role in balancing interactions between text guidance and visual generation, ensuring coherent and stable outputs. For more detailed analysis, please refer to **Appendix** §A.

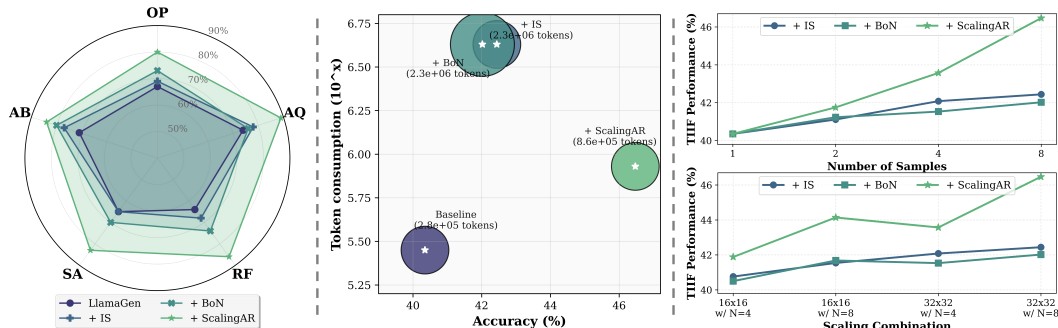

Figure 5: (*Left*) User study across five dimensions: overall preference, aesthetic quality, realism fidelity, semantic alignment, attribute binding. (*Middle*) Visual token consumption of `ScalingAR` *vs.* baselines on TIIF-Bench. (*Right*) Scaling width and depth across sample number and token length.

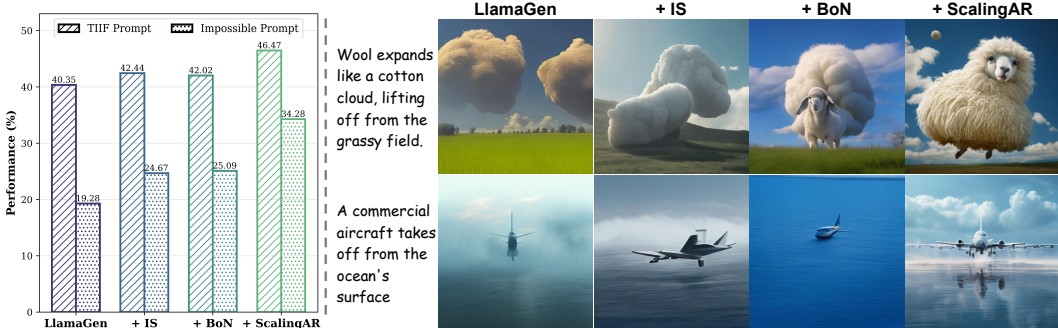

Figure 6: Robustness testing with impossible prompt. Detailed prompts are provided in **Appendix** §A.

## 5.4 SCALABILITY & ROBUSTNESS ANALYSIS

To answer **RQ4**, we compare `ScalingAR` with other TTS strategies (*i.e.*, IS and BoN) in scaling width (*i.e.*, sample number $N$) and depth (*i.e.*, token length), as shown in Figure 5 (*Right*). To further assess the robustness of `ScalingAR`, we adopt the idea of "impossible prompting" (Bai et al., 2025) (*e.g.*, "A young boy ... using chopsticks as a writing instrument, ... in a photo-realistic scene...") to evaluate its performance even when none of the candidates are ideal, with the results presented in Figure 6. Our observations are summarized as follows: **Obs.⑤** `ScalingAR` *unlocks scalable generalization across both width and depth.* As shown in Figure 5 (*Right*), `ScalingAR` consistently outperforms IS and BoN across varying sample numbers and token lengths. This suggests that our scaling strategy enables performance to scale up effectively as scaling width and depth increase, making it a reliable solution for diverse autoregressive tasks. **Obs.⑥** `ScalingAR` *empowers robust generation beyond standard scenarios.* Figure 6 (*Left*) demonstrates that under impossible prompts for unrealistic scenarios, `ScalingAR` exhibits clear robustness advantages over baselines. Furthermore, Figure 6 (*Right*) confirms that our method achieves more effective scaling when generating under challenging conditions, highlighting its adaptability and reliability in adverse scenarios.

## 6 CONCLUSION

In this work, we introduce `ScalingAR`, the first test-time scaling framework tailored to next-token prediction autoregressive image generation. Unlike existing TTS strategies, `ScalingAR` proposes to explore visual token entropy for the first time as intrinsic signals, without relying on partial decoding or external rewards. By adopting a two-level design: Profile Level for calibrated confidence profiling and Policy Level for adaptive pruning and dynamic conditioning, `ScalingAR` achieves phase-aware control, enhancing generation quality with minimal additional token consumption. Comprehensive evaluations on both general and compositional capability benchmarks demonstrate that `ScalingAR` substantially improves the generation quality of existing AR models, along with generalizability and robustness, making it a strong baseline for AR image generation test-time scaling.

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

# A MORE EXPERIMENTAL SETTINGS AND ANALYSIS

## A.1 MORE DETAILS OF EXPERIMENTAL SETTINGS

**Implementation Details**   We implement our `ScalingAR` and conduct all experiments on NVIDIA H100 GPUs. Here we detail the hyperparameters.

| Notation | Definition | Value |
|---|---|---|
| $\lambda_{\text{tok}}$ | token confidence smoothing factor | 0.2 |
| $\alpha_H/\alpha_M$ | token-level confidence weights | 0.5/0.5 |
| $w_{\text{tok}}$ | token confidence weight | 0.65 |
| $w_{\text{blk}}$ | worst-block stability weight | 0.35 |
| $\lambda_I$ | smoothing factor for intrinsic channel score | 0.2 |
| $b$ | block size for spatial entropy | 4 |
| $\rho_{\min}$ | minimum fill ratio for spatial entropy | 4 |
| $q$ | worst-$q$% subset size | 0.1 |
| $w_I$ | intrinsic channel weight | 0.75 |
| $w_D$ | conditional channel weight | 0.25 |
| $y_{\text{sigmoid}}$ | affine-sigmoid calibration | 1.0 |
| $W_0$ | warm-up period | 12.5% |
| $\mathbf{p}$ | confidence threshold quantile | 0.2 |
| $\lambda_\theta$ | EMA update rate for threshold | 0.2 |
| $\Delta_{\text{rec}}$ | recovery window | 32 |
| $\delta_{\text{rec}}$ | recovery threshold | 0.05 |
| $T_{\min}$ | protection horizon | 5% |
| $C_{\text{hard}}$ | hard-fail confidence guard | 0.3 |
| $\alpha$ | influence coefficient for condition utilization | 0.3 |
| $\beta$ | influence coefficient for intrinsic volatility | 0.4 |
| $\lambda$ | influence coefficient for rebound | 0.4 |

**Captions of Figure 1**   For qualitative results in Figure 1 (*Top*), we further detail the prompts here:

- **1st**: *"A red rose in full bloom sits on the top, above a pink rosebud."*

- **2nd**: *"A photo of a cute puppy playing in a sunny backyard."*

- **3rd**: *"A young boy holding a mysterious key, embarking on an adventure through various landscapes to find hidden treasure."*

- **4th**: *"A masked hero jumping from a rooftop, comic book style with bold outlines and dialogue bubbles."*

- **5th**: *"A close-up of an anime woman's face with a shocked expression, featuring dark hair, drawn in the anime style. The image showcases colorful animation stills, close-up intensity, soft lighting, a low-angle camera view, and high detail."*

**Robustness Testing**   To evaluate the robustness of `ScalingAR`, we further employ prompts from IPV-TXT from Impossible Videos [ICML'25] (Bai et al., 2025). Specifically, we filtered prompts suitable for image generation from IPV-TXT, then employed Impossible Prompt Following (IPF) as the evaluation metric, which measures the alignment between generated images and the semantic intent of impossible prompts. Following Bai et al. (2025), we employed GPT-4o to perform binary judgments on each image based on prompt adherence. For qualitative results in Figure 6 (*Right*):

- **1st**: *"A sheep peacefully grazing in a realistic meadow suddenly defies gravity as its wool expands dramatically, causing its body to balloon up like a cotton cloud. The fluffy animal then lifts off from the grassy field and drifts upward into the blue sky, its transformed woolly coat acting like a natural balloon."*

- **2nd**: *"A commercial aircraft inexplicably takes off from the ocean's surface as if the water were a solid runway, defying physics in this photo-realistic scene. The calm, glassy sea appears to have transformed into a firm platform, allowing the plane to accelerate and lift off smoothly, with spray trailing behind its wheels like it would on a wet tarmac."*

**User Study** We conducted a user study to evaluate human preferences using the mean opinion score (MOS) metric. We designed a user-friendly interface to facilitate the evaluation process and collected feedback from a total of 15 volunteer participants. The detailed instructions provided to the participants are as follows:

---

**User Study: Autoregressive Image Generation**

Thank you for participating in our user study! Please follow these steps to complete your evaluation:

1. **Image Generation:** Carefully read the target prompt provided, and then view the provided images.
2. **Scoring Criteria:** Assign a score to each generated image based on the following aspects (1 being the lowest, 5 being the highest):

- *Overall Quality*: The overall perceived quality and appeal of the generated image.
- *Aesthetic Quality*: The visual aesthetics, composition, and artistic merit of the image.
- *Realism Fidelity*: How realistically and faithfully the image captures the intended scene or subject matter.
- *Semantic Alignment*: How well the generated image aligns with and represents the meaning of the textual prompt.
- *Attribute Binding*: The degree to which the image accurately depicts the specific attributes and details described in the text.

3. **Submission:** Click the "Submit Scores" button to submit your scores.

**Notations:**

1. We observe that the edge browser is not fully compatible with our interface. Chrome is recommended.
2. Remember to click the "Submit Scores" button after your evaluation.
3. If you see that images and the score sliders are not aligned, shrinking your page usually works.
4. If the page is not responsive for a long time, please try to refresh it.
5. If you have any questions, please directly ping us. Thank you for your time and effort!

---

## A.2 MORE ANALYSIS

**Analysis of Global Confidence & Guidance Weights** Figure 7 presents a detailed analysis of the impact of weights of unified confidence and guidance scheduler on the performance of `ScalingAR` on the TIIF-Bench. ❶ **Unified Confidence** (Figure 7 (*Top*)): Varying the balance between the Intrinsic ($w_I$) and Conditional ($w_D$) channels shows that emphasizing the Intrinsic channel slightly ($w_I/w_D = 0.75/0.25$) achieves the best TIIF-Bench performance across all subsets. This highlights the importance of capturing local uncertainty and stability while maintaining semantic alignment. Omitting the Conditional Channel (1.00/0.00) degrades performance, confirming its complementary role. ❷ **Guidance Scheduler** (Figure 7 (*Bottom*)): Adjusting the weights $\alpha$, $\beta$, and $\lambda$, which control conditional utilization, intrinsic volatility, and confidence rebound, respectively, reveals that moder-

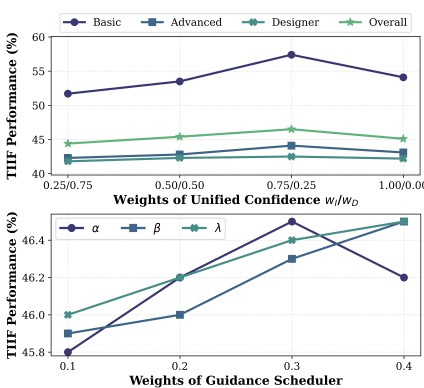

Figure 7: Analysis of `ScalingAR` for weights of Unified Confidence (***Top***) and Guidance Scheduler (***Bottom***).

ate emphasis on intrinsic volatility and rebound ($\beta, \lambda$) improves performance. The weight $\alpha$ peaks at 0.3, suggesting overemphasis may reduce diversity. This confirms the need for balanced, dynamic guidance to optimize semantic fidelity and diversity.

**Analysis of Adaptive Termination Gate** We further analyze the impact of the confidence threshold quantile **p** and the recovery threshold $\delta_{\mathrm{rec}}$ on the performance and token efficiency of `ScalingAR`, as illustrated in Figure 8. ❶ **Confidence Threshold** (Figure 8 (*Left*)): The choice of confidence threshold critically balances pruning aggressiveness and generation quality. Setting **p** too low leads to insufficient pruning, resulting in higher token consumption with limited accuracy gains. Conversely,

an overly high threshold causes premature termination of promising trajectories, degrading accuracy despite lower token usage. Our experiments show that an intermediate threshold (*e.g.*, $p = 0.20$) achieves the best trade-off, significantly improving accuracy while maintaining efficient token consumption compared to both baseline and extreme settings. ❷ **Recovery Threshold** (Figure 8 (*Right*)): The recovery mechanism safeguards

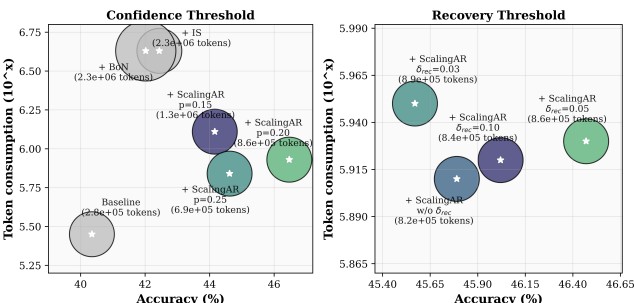

Figure 8: Analysis of `ScalingAR` for thresholds of Confidence (*Left*) and Recovery (*Right*).

against false positives by allowing trajectories to rebound from transient confidence dips. Disabling this mechanism leads to noticeable performance drops, highlighting its necessity. Furthermore, setting the recovery threshold $\delta_{rec}$ too low or too high adversely affects accuracy and efficiency: a low threshold permits premature recovery of poor trajectories, increasing token cost, while a high threshold delays recovery, risking early termination of viable samples. An optimal value (*e.g.*, $\delta_{rec} = 0.05$) balances these effects, maximizing accuracy with minimal token overhead.

**Analysis of Ablation on Policy Level** While ablation study (Table 2) in main text focuses on the Profile Level, we conducted additional ablation studies to evaluate the contributions of the Policy Level, which builds upon the Profile Level, as shown in Table 3. (*i*) The "Termination Only" setup improves performance across all metrics,

Table 3: Ablation of Policy Level.

| Method | Bas.↑ | Adv.↑ | Des.↑ | Overall↑ |
|---|---|---|---|---|
| LlamaGen | 49.6 | 40.4 | 40.3 | 40.4 |
| + Termination Only | 54.1 | 43.1 | 42.2 | 45.1 |
| + Scheduler Only | 53.6 | 42.0 | 41.0 | 43.8 |
| + `ScalingAR` (**Ours**) | **57.4** | **44.1** | **42.5** | **46.5** |

highlighting its ability to prune low-confidence trajectories and mitigate failure modes, ensuring stable generation. (*ii*) The "Scheduler Only" setup also yields notable gains, demonstrating its effectiveness in dynamically modulating conditioning strength to balance semantic alignment and diversity. (*iii*) Integrating both mechanisms achieves the best results, showing their complementary roles in improving generation quality and efficiency. These results validate the Policy Level as essential for enhancing autoregressive image generation.

Table 4: Computation consumption comparison on GenEval with NVIDIA 140G H200 GPU.

| Method | $N$ | Per-step WC (s) | Overall WC (s) | Matched Tokens/Img | FLOPs (TFLOPs) | Memory (GB) | Performance |
|---|---|---|---|---|---|---|---|
| LlamaGen | 1 | 0.024 | 24.93 | 1024 | 5.60 | 6.44 | 0.32 |
| + BoN | 8 | 0.025 | 218.44 | 8192 | 39.12 | 48.72 | 0.15 |
| + `ScalingAR` (**Ours**) | 8 | 0.029 | 69.56 | 2350 | 4.23 | 18.16 | 0.36 |

**Analysis of `ScalingAR`'s Efficiency** We conclude the average computation consumption of `ScalingAR` in Table 4.

**Analysis of Local Confidence Weights** We further analyze the impact of the weighting strategies for Token-Level Confidence $\alpha_H/\alpha_M$ and Worst-Block Stability $w_{tok}/w_{blk}$ on the performance of `ScalingAR`, as illustrated in Figure 9. ❶ **Token-Level Confidence Weights** (Figure 9 (*Left*)): Adjusting the balance between entropy-based uncertainty ($\alpha_H$) and margin-based confidence

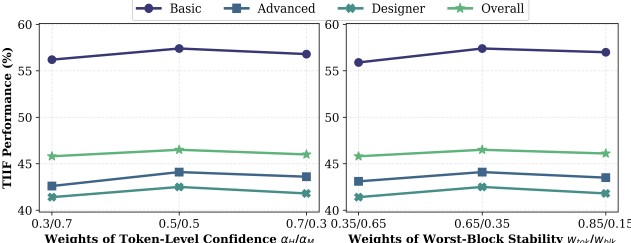

Figure 9: Analysis of `ScalingAR` for weights of Token-Level Confidence (*Left*) and Worst-Block Stability (*Right*).

($\alpha_M$) reveals that prioritizing entropy signals ($\alpha_H/\alpha_M = 0.7/0.3$) achieves the best overall performance across all metrics. This suggests that entropy provides a more robust signal for capturing localized instability during generation. Conversely, overemphasizing margin-based confidence ($\alpha_H/\alpha_M = 0.3/0.7$) leads to performance degradation, particularly in advanced and designer subsets, as it fails to fully capture nuanced instability patterns. A balanced setting ($\alpha_H/\alpha_M = 0.5/0.5$) offers a reasonable trade-off, though slightly underperforms the optimal configuration. ❷ **Worst-Block Stability Weights** (Figure 9 (*Right*)): Varying the balance between token-level confidence ($w_{tok}$) and block-level stability ($w_{blk}$) shows that an emphasis on token-level signals ($w_{tok}/w_{blk} = 0.85/0.15$) slightly reduces performance, particularly in the advanced and

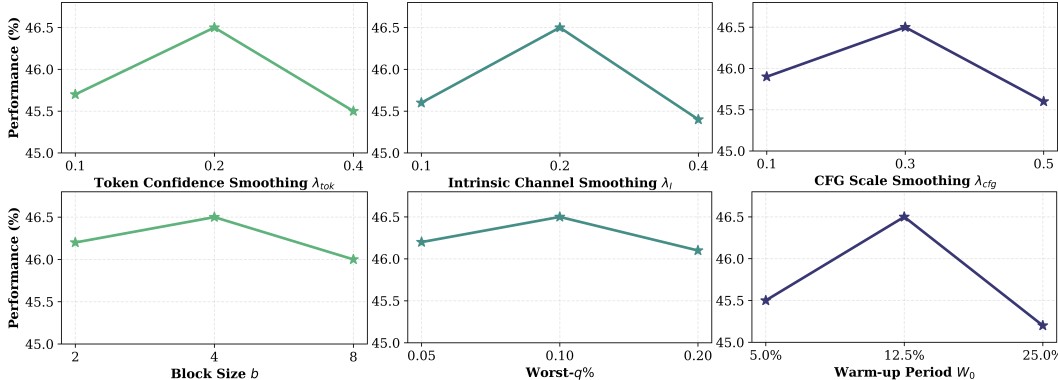

Figure 10: Analysis of hyperparameters. (**a**) Token Confidence Smoothing. (**b**) Intrinsic Channel Smoothing. (**c**) CFG Scale Smoothing. (**d**) Block Size. (**e**) Worst-$q\%$. (**f**) Warm-up Period.

designer subsets, as it underweights spatial anomalies that propagate into global failures. On the other hand, overemphasizing block-level stability ($w_{\text{tok}}/w_{\text{blk}} = 0.35/0.65$) also degrades results, as it may overreact to localized noise. The optimal configuration ($w_{\text{tok}}/w_{\text{blk}} = 0.65/0.35$) balances token-level and block-level signals effectively, achieving the highest scores across most metrics.

**Analysis of Smoothing Factors** ❶ **Token Confidence Smoothing ($\lambda_{\text{tok}}$):** As shown in Figure 10 (*a*), the choice of $\lambda_{\text{tok}}$ significantly impacts the performance of `ScalingAR`. A moderate smoothing factor ($\lambda_{\text{tok}} = 0.2$) achieves the best performance across all subsets, as it effectively balances stability and responsiveness in token-level confidence signals. Setting $\lambda_{\text{tok}}$ too low ($\lambda_{\text{tok}} = 0.1$) results in noisy signals, while overly high smoothing ($\lambda_{\text{tok}} = 0.4$) delays the system's adaptability to dynamic changes, degrading performance. ❷ **Intrinsic Channel Smoothing ($\lambda_I$):** Figure 10 (*b*) demonstrates that $\lambda_I = 0.2$ provides the best TIIF performance. Lower values ($\lambda_I = 0.1$) fail to stabilize the intrinsic confidence signal, leading to suboptimal trajectory pruning. On the other hand, higher values ($\lambda_I = 0.4$) overly smooth the signal, reducing sensitivity to localized instability and resulting in degraded generation quality. ❸ **CFG Scale Smoothing ($\lambda_{\text{cfg}}$):** In Figure 10 (*c*), the performance peaks at $\lambda_{\text{cfg}} = 0.3$, reflecting an optimal trade-off between smooth transitions in CFG scale adjustments and responsiveness to real-time confidence signals. Smaller values ($\lambda_{\text{cfg}} = 0.1$) introduce excessive fluctuations, while larger values ($\lambda_{\text{cfg}} = 0.5$) hinder the system's ability to adapt to changing confidence states.

**Analysis of Spatial Entropy** ❶ **Block Size ($b$):** As illustrated in Figure 10 (*d*), a block size of $b = 4$ achieves the best performance. Smaller blocks ($b = 2$) are overly sensitive to local noise, leading to false positives in detecting instability. Conversely, larger blocks ($b = 8$) fail to capture fine-grained spatial anomalies, resulting in reduced effectiveness in trajectory pruning. ❷ **Worst-$q\%$:** Figure 10 (*e*) shows that setting $q = 0.10$ yields the highest performance. A smaller $q$ ($q = 0.05$) underestimates the impact of localized high-entropy regions, while a larger $q$ ($q = 0.20$) dilutes the focus on the most problematic areas, reducing the precision of the stability signal.

**Analysis of Warm-up Period** Figure 10 (*f*) highlights the importance of an appropriate warm-up period. The best performance is achieved with $W_0 = 12.5\%$, which provides sufficient time for confidence signals to stabilize before applying trajectory pruning. A shorter warm-up period ($W_0 = 5.0\%$) leads to premature pruning of promising trajectories, while a longer warm-up period ($W_0 = 25.0\%$) delays intervention, reducing efficiency and quality.

# B  RESULTS OF MORE BASE MODELS

To further validate the generalizability of `ScalingAR`, we deployed our method on two additional AR models: SimpleAR-1.5B (Wang et al., 2025a) and Janus-Pro-1B (Chen et al., 2025a). Importantly, the hyperparameter settings for `ScalingAR` were kept consistent with those used in the main experiments on LlamaGen and AR-GRPO, without any model-specific tuning. This ensures a fair evaluation of `ScalingAR`'s adaptability across different architectures and scales. Quantitative

Table 5: Evaluation of `ScalingAR` on more base models on GenEval.

| Method | TO↑ | Pos.↑ | CA↑ | Overall↑ |
|---|---|---|---|---|
| SimpleAR | 0.90 | 0.28 | 0.45 | 0.63 |
| + `ScalingAR` (**Ours**) | **0.93** | **0.36** | **0.51** | **0.67** |
| Janus-Pro | 0.82 | 0.65 | 0.56 | 0.73 |
| + `ScalingAR` (**Ours**) | **0.87** | **0.69** | **0.61** | **0.77** |

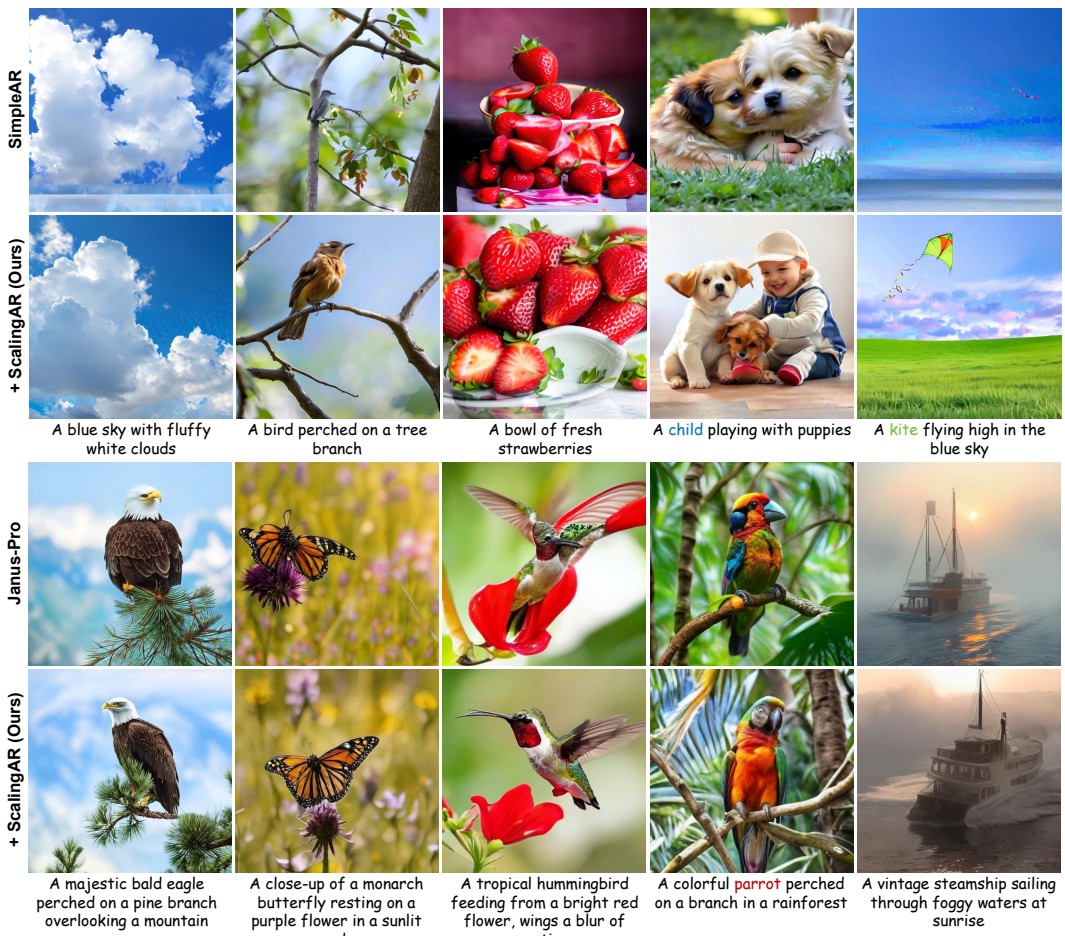

Figure 11: Qualitative results of `ScalingAR` on SimpleAR (*Top*) and Janus-Pro (*Bottom*).

results in Table 5 and qualitative results in Figure 11 show significant performance improvements for both models, demonstrating `ScalingAR`'s effectiveness and broad applicability as a general-purpose stabilization framework.

## C    FURTHER ILLUSTRATION OF ENTROPY IN AR IMAGE GENERATION

A key motivation behind our `ScalingAR` lies in the observation that high-entropy/low-confidence regions often exhibit greater uncertainty, which increases the likelihood of undesirable outcomes. While high entropy *does not* guarantee poor results, it correlates strongly with elevated error probabilities, making it a critical signal for stabilizing AR image generation.

**Relevant Evidence**    Similar observations have been validated across various domains: ❶ Entropy calibration in language models: Cao et al. (2025) demonstrated that high local token entropy correlates with higher error probabilities, highlighting its role as a risk indicator in generative tasks. ❷ Reinforcement learning mechanisms for reasoning: Works (Cui et al., 2025; Fu et al., 2025; Wang et al., 2025b) for LLM Reasoning treat high-entropy tokens as positions with dense information but unstable decisions or higher error risks. These findings underscore the necessity of carefully managing entropy during generation to balance exploration and stability.

**Connection on `ScalingAR`**    Our method, `ScalingAR`, can be interpreted as a stabilization mechanism that prunes trajectories with low confidence, effectively mitigating the risks associated with high-entropy regions. By focusing on confidence signals, `ScalingAR` ensures that the generation process avoids prolonged instability, leading to improved image quality. In Figure 1 (*Bottom Left*) and Figure 3 (*Left*), we compare the confidence distributions of `ScalingAR` and the base model.

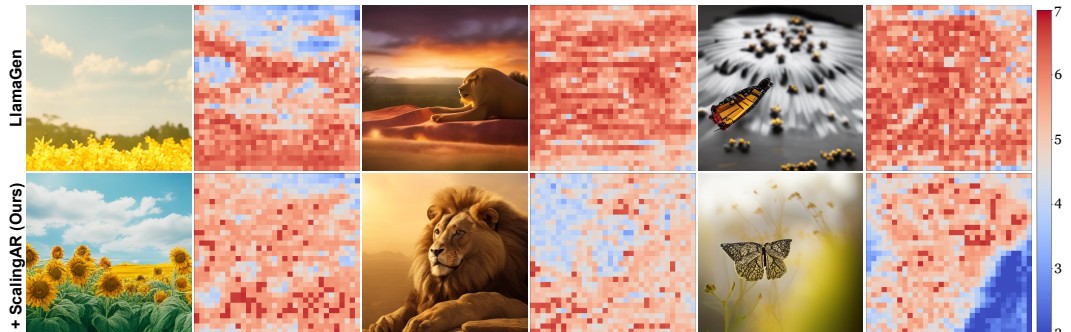

Figure 12: Visualization of token entropy. (*1st*) *A sunflower field stretching to the horizon under a bright blue sky.* (*2nd*) *A majestic lion resting on a rocky outcrop in the golden savanna light.* (*3rd*) *A detailed macro shot of a butterfly on a blooming flower.*

The results clearly show that higher token confidence correlates with better image quality, further validating our motivation to leverage confidence signals for trajectory pruning.

**Visualizing Token Entropy in Generated Images**   To provide a more intuitive understanding, we visualize the entropy distributions of generated images in Figure 12. The figure highlights that regions with poor generation quality often correspond to higher entropy, reinforcing the notion that high-entropy tokens are more likely to contribute to undesirable outcomes. `ScalingAR`'s ability to suppress these regions through confidence-based pruning plays a pivotal role in achieving stable and high-quality image generation.

## D ALGORITHM WORKFLOW

We conclude the overall algorithm workflow of `ScalingAR` in Algorithm 1.

---

**Algorithm 1: `ScalingAR` Workflow**

---

**Input:** Prompt $y$, AR model $p_\theta(\cdot \mid \cdot)$, CFG scale $g$, pruning threshold $\tau$, max steps $T$, beam width $N$, top-$k$ $k$
**Output:** Final image $\hat{x}$

1  Initialize candidate set $\mathcal{S}_0 \leftarrow \{(\text{seq} = \emptyset)\}_{i=1}^N$
2  **for** $t \leftarrow 1$ **to** $T$ **do**
3      **if** $\mathcal{S}_{t-1}$ *is empty* **then**
4          break
5      **end**
6      **foreach** $s \in \mathcal{S}_{t-1}$ **do**
7          Compute conditional logits $\ell_c \leftarrow p_\theta(\cdot \mid s.\text{seq}, y)$
8          Compute unconditional logits $\ell_u \leftarrow p_\theta(\cdot \mid s.\text{seq}, \varnothing)$
9          Guided logits $\tilde{\ell} \leftarrow \ell_u + g \cdot (\ell_c - \ell_u)$
10         Compute probs $p_{\text{tok}} \leftarrow \text{softmax}(\tilde{\ell})$
11         Sample token $x_t \sim p_{\text{tok}}$
12         Append $x_t$ to $s.\text{seq}$
13         Compute entropy $H_t \leftarrow -\sum_j p_{\text{tok}}(j) \log p_{\text{tok}}(j)$
14         Compute confidence $C_t \leftarrow -\frac{1}{k} \sum_{j \in \text{Top-}k} \log p_{\text{tok}}(j)$
15         Compute utilization $U_t \leftarrow \text{KL}(\text{softmax}(\ell_c) \| \text{softmax}(\ell_u))$
16         Compute fused confidence $\Phi_t \leftarrow w_c C_t + w_u U_t$
17     **end**
18     Compute threshold $\tau_t \leftarrow p$-quantile of $\{\Phi_t(s)\}$
19     Prune candidates with $\Phi_t(s) < \tau_t$
20     $\mathcal{S}_t \leftarrow$ survivors after pruning
21 **end**
22 Select best candidate $\hat{s} \leftarrow \arg\max_{s \in \mathcal{S}_t} \Phi_t(s)$
23 Decode $\hat{s}$ to image $\hat{x}$ and **return** $\hat{x}$

---

## E    EXHIBITION BOARD

We provide more comparison results here in Figure 13 on AR-GRPO and Figure 14 on LlamaGen.

## F    LIMITATION AND FUTURE WORKS

`ScalingAR` pioneers test-time scaling for autoregressive image generation but faces key challenges. AR image modeling involves complex dependencies, making confidence estimation difficult; our exploration of token entropy is a first step but may not fully capture uncertainty and semantic alignment. Additionally, the approach relies on model calibration and entropy signals, which can vary with training and architecture. Future work includes developing finer-grained confidence measures for more precise scaling, and integrating entropy-based signals into both training-time and test-time to create a more unified pipeline.

## G    THE USE OF LLMs

This research does not involve LLMs in terms of training or fine-tuning as part of its core contributions. The use of LLMs is limited to polishing the writing of the manuscript. These uses do not impact the originality or core methodology of the research, and therefore do not require detailed declaration.

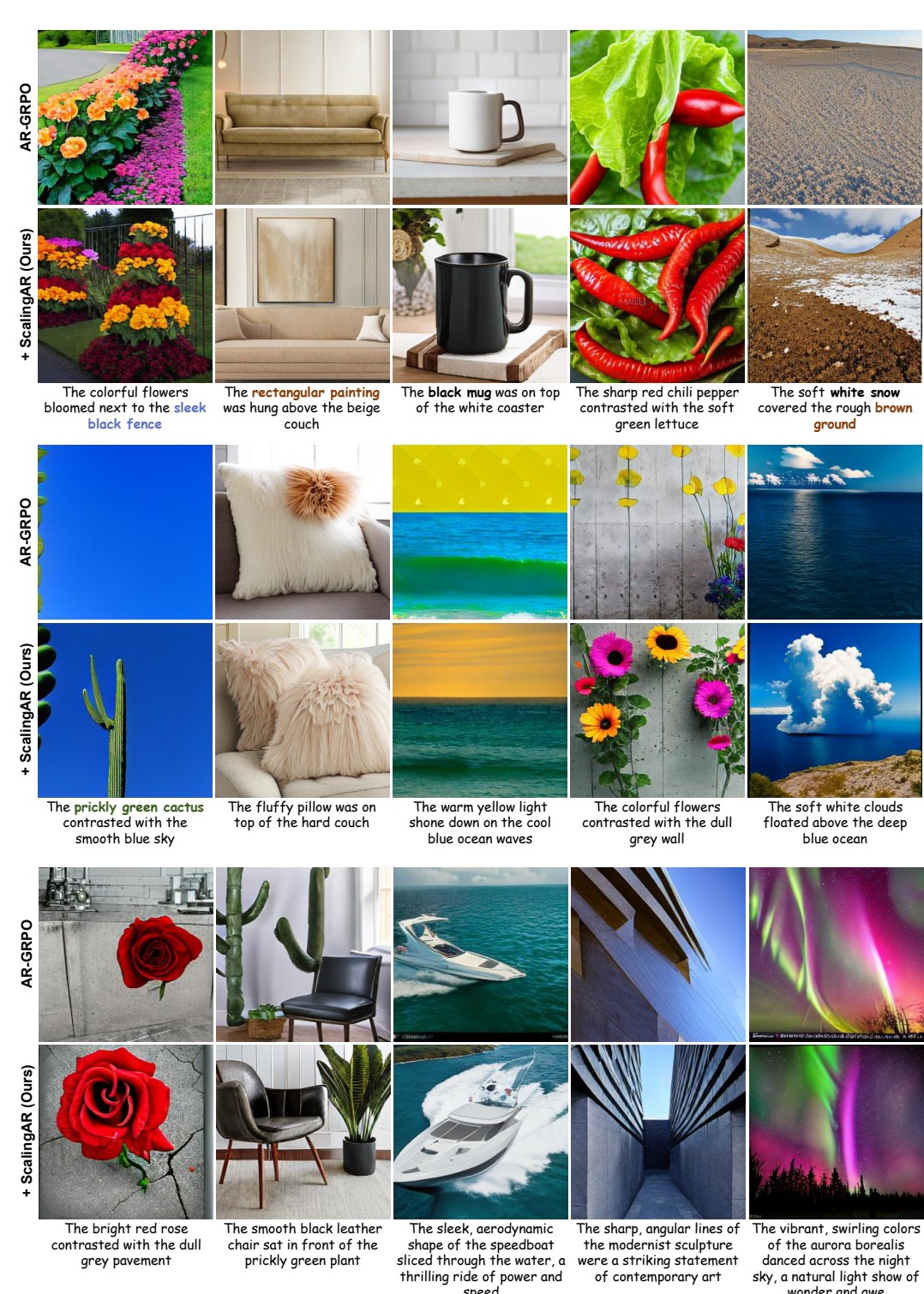

Figure 13: More results demonstrations of `ScalingAR` on AR-GRPO (Yuan et al., 2025).

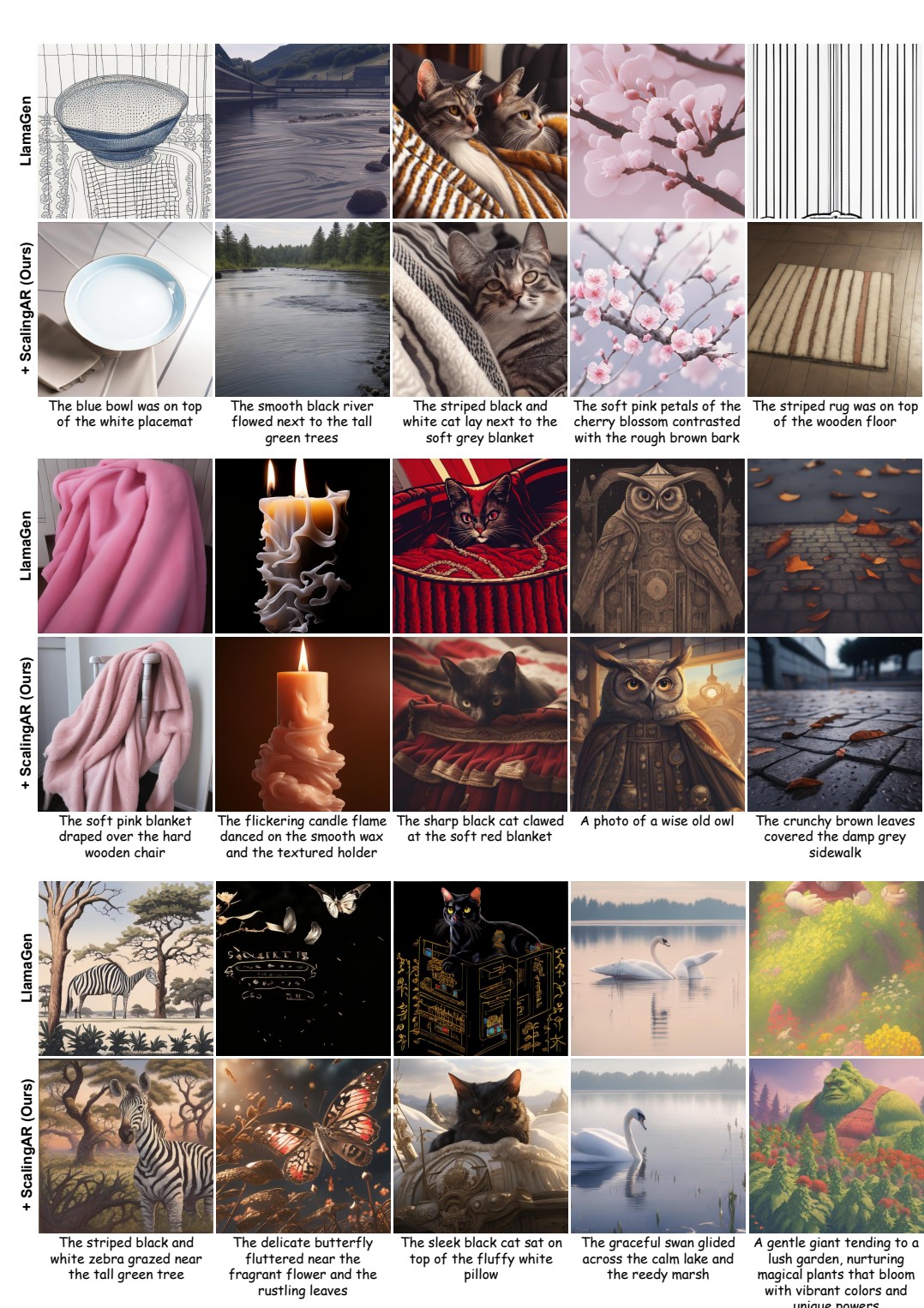

Figure 14: More results demonstrations of `ScalingAR` on LlamaGen (Sun et al., 2024).

