# OpenReview forum: "Go with Your Gut: Scaling Confidence for Autoregressive Image Generation"
_ICLR.cc/2026/Conference — Submitted to ICLR 2026_

### Official Review · Reviewer_E86T · 2025-10-27

**Soundness:** 2
**Presentation:** 3
**Contribution:** 2
**Rating:** 4
**Confidence:** 3

**Summary:**

The paper introduces a test-time controller for autoregressive image models that reads the model’s logits, not partial renders, to decide how to spend compute. It builds a two-channel confidence: an intrinsic stability signal from token entropy and a worst-block grid score, and a conditional-utilization signal from KL between conditional and unconditional logits to catch semantic fade. A lightweight policy prunes low-promise runs with a recovery check, and schedules classifier-free guidance on the fly, raising guidance when utilization drops and relaxing it once predictions stabilize. On GenEval and TIIF, it lifts strong AR bases and beats IS or BoN with fewer tokens, so you get better quality per unit test compute.

**Strengths:**

1. Uses logit confidence instead of partial renders. That targets the real pain in AR images: intermediate images are costly and uninformative, logits are cheap and continuous.
2. The story reads straight. One state, confidence from logits; one purpose, allocate compute; two actions, prune and schedule guidance.
3. Well-scoped claim. Improves quality per unit test compute, not a new training objective, not a reward model; scope and evidence match.

**Weaknesses:**

1. Too many hyperparameters: weights, windows, percentiles, rebound length, CFG rates, clamps.
2. Many moving parts: worst-block, conditional channel, CFG, rebound, threshold.
3. Hand-designed controller. Not learned, no optimality claim.
4. Compute-normalized fairness. Account for double forward passes for conditional and unconditional, edit scheduling, and any candidate scoring overheads.
5. Confidence, entropy-based pruning and CFG scheduling exist in nearby areas. The novelty is the logit-only, image-specific fusion.

**Questions:**

1. sensitivity analysis of the hyper parameters
2. component ablations
3. report success under normalized compute: matched token, FLOP, and memory budgets.

---

> ### Author Response · Authors · 2025-11-20
> **[Part 1/2] Response to Reviewer E86T**
>
> Thank you for your insightful comments and questions on our manuscript! We have carefully considered each point and have made the necessary revisions and clarifications to address your concerns:
>
> ---
> > `Weakness 1 & Question 1`: Too many hyperparameters: weights, windows, percentiles, rebound length, CFG rates, clamps.
>
> Thank you for highlighting this critical point! We acknowledge that our methodology involves several hyperparameters, which were carefully designed to address the challenges of AR image generation. Through our early exploration, **we found these parameters to be highly effective, and our intent is to provide the community with actionable insights for future work.**
>
> We apologize for only including analysis of the core hyperparameters in the initial version of the manuscript. Following your suggestion, we have now expanded the sensitivity analysis in `Appendix A.2` to include additional parameters such as **token-level confidence/worst block stability weights**, **smoothing factors**, **recovery window**, and **warm-up period**.
>
> We hope this additional analysis addresses your concern and provides greater clarity on the role of these parameters. Thank you again for your valuable feedback!
>
> ---
> > `Weakness 2`: Many moving parts: worst-block, conditional channel, CFG, rebound, threshold.
>
> Thank you for your comment! While ScalingAR involves multiple components, they work together to address key challenges in AR image generation. Importantly, ScalingAR uses the **same set of parameters across different base models without requiring model-specific tuning**.
>
> To further demonstrate its robustness, we evaluated ScalingAR on additional base models (*i.e.*, SimpleAR-1.5B [1] and Janus-Pro-1B [2]), and the results confirm its effectiveness:
>
> *Table A: Evaluation of ScalingAR on more base models on GenEval.*
> ||Two Object|Position|Color Attribute|Overall|
> |-|-|-|-|-|
> |SimpleAR [1]|0.90|0.28|0.45|0.63|
> |**+ ScalingAR (Ours)**|**0.93**|**0.36**|**0.51**|**0.67**|
> |Janus-Pro [2]|0.82|0.65|0.56|0.73|
> |**+ ScalingAR (Ours)**|**0.87**|**0.69**|**0.61**|**0.77**|
>
> These results demonstrate that ScalingAR consistently improves performance across diverse architectures without additional tuning. Qualitative results are provided in `Appendix B`.
>
> ---
> > `Weakness 3`: Hand-designed controller. Not learned, no optimality claim.
>
> Thank you for your valuable feedback. We would like to humbly clarify that our controller leverages **intrinsic signals** already present in AR image generation models, such as confidence scores and condition usage rates. These signals can be directly extracted **without requiring additional training or learning processes**.
>
> This design ensures that our controller is lightweight, computationally efficient, and easy to integrate into existing AR models, aligning with our goal of providing a practical and effective test-time scaling framework. While we acknowledge the potential benefits of learning-based controllers, we believe our approach achieves a favorable balance between simplicity, efficiency, and performance.
>
> ---
> > `Weakness 4 & Question 3`: Report success under normalized compute: matched token, FLOP, and memory budgets.
>
> Thank you for bringing up this important point! To address your concern, we supplement our analysis with a detailed comparison of computation costs on an NVIDIA 140G H200 GPU, as shown below:
>
> *Table B: Computation consumption comparison on GenEval.*
> ||$N$|Avg. Per-step Time (s)|Avg. Overall Time (s)|Token Consumption/Img|FLOPs (TFLOPs)|Memory (GB)|Performance|
> |-|-|-|-|-|-|-|-|
> |LlamaGen|1|0.024|24.93|1024|5.60|6.44|0.32|
> |+ BoN|8|0.025|218.44|8192|39.12|48.72|0.15|
> |**+ ScalingAR (Ours)**|8|0.029|69.56|2350|4.23|18.16|0.36|
>
> As the results show, ScalingAR achieves significant performance improvements while requiring relatively modest additional computation compared to baseline strategies. This aligns with the conclusions in our main text and further highlights the practicality of our approach.

---

> ### Author Response · Authors · 2025-11-20
> **[Part 2/2] Response to Reviewer E86T**
>
> > `Weakness 5`: Confidence, entropy-based pruning and CFG scheduling exist in nearby areas. The novelty is the logit-only, image-specific fusion.
>
> We sincerely appreciate your thoughtful feedback! While we agree that confidence cues, entropy-based pruning, and CFG scheduling exist in nearby areas, our key contribution lies in **making these mechanisms actionable for NTP AR image generation** through a **logit-only, image-specific fusion**, which operates without intermediate renders or external reward models:
>
> - **Spatial–temporal profiling on logits:** Worst-block stability and rebound mechanisms detect persistent local instability in the 2D token grid, which is absent in text-based routing.
> - **Dual-channel fusion:** Combines entropy/margin for intrinsic stability with KL divergence $\mathrm{KL}(p_c\parallel p_u)$ for prompt utilization, disentangling stability and conditioning across decoding phases.
> - **Policy coupling:** The fused state drives early termination with safeguards and phase-aware CFG scheduling, leveraging logits already computed (no additional forward passes required).
>
> We hope this clarification sufficiently addresses your concerns and highlights the novelty of our approach.
>
> ---
> > `Question 2`: Ablations of the two components: which plays the role?
>
> In the ablation study in `Table 2`, we demonstrated the step-by-step performance impact of removing Profile Level components. Since the Policy Level builds on the Profile Level, our main text focused on ablations at the Profile Level (*e.g.*, "w/o Conditional Channel" corresponds to "Termination Only"). To further clarify the role of each component, we now provide a more detailed ablation of the Profile Level and supplement it with an ablation of the Policy Level:
>
> *Table C: Step-by-step abalation analysis on Profile Level on TIIF-Bench.*
> ||Basic|Advanced|Designer|Overall|
> |-|-|-|-|-|
> |**+ ScalingAR (Ours)**|**57.4**|**44.1**|**42.5**|**46.5**|
> |w/o Conditional Channel|54.1|43.1|42.2|45.1|
> |w/o Worst-Block Stability|52.3|41.8|41.4|44.2|
> |w/o Token-Level Confidence|49.6|40.4|40.3|40.4|
>
> *Table D: Ablation analysis on Policy Level on TIIF-Bench.*
> ||Basic|Advanced|Designer|Overall|
> |-|-|-|-|-|
> |LlamaGen|49.6|40.4|40.3|40.4|
> |+ Termination Only|54.1|43.1|42.2|45.1|
> |+ Scheduler Only|53.6|42.0|41.0|43.8|
> |**+ ScalingAR (Ours)**|**57.4**|**44.1**|**42.5**|**46.5**|
>
> These results show that each component of ScalingAR contributes to its overall performance, with both the Profile and Policy Levels playing distinct but complementary roles. We hope this detailed analysis addresses your concern. Thank you again for your valuable feedback!
>
> ---
> [1] SimpleAR: Pushing the Frontier of Autoregressive Visual Generation through Pretraining, SFT, and RL
>
> [2] Janus-Pro: Unified Multimodal Understanding and Generation with Data and Model Scaling

---

> ### Author Response · Authors · 2025-11-25
> **Thank you & Looking forward to further discussion**
>
> Dear Reviewer E86T,
>
> We deeply appreciate your thoughtful feedback and the opportunity to address your concerns. Recognizing the existing discrepancies in the understanding of our manuscript, we have outlined your key concerns and our responses for enhanced communication:
>
> 1. **Too many hyperparameters `Weakness 1 & Question 1`** We expand the sensitivity analysis in `Appendix A.2` to include additional hyperparameters (*e.g.*, token-level confidence, smoothing factors, recovery window) and clarify their roles in improving AR image generation.
> 2. **Many moving parts in the methodology `Weakness 2`** While ScalingAR involves multiple components, we demonstrate its robustness by applying the same parameters across different models (*e.g.*, SimpleAR-1.5B, Janus-Pro-1B). Results highlight consistent performance improvements without additional tuning.
> 3. **Hand-designed controller `Weakness 3`** We clarify that our lightweight controller leverages intrinsic signals (*e.g.*, confidence scores) without requiring additional training, achieving a balance between simplicity, efficiency, and performance.
> 4. **Normalized compute comparison `Weakness 4 & Question 3`** We provide a detailed analysis of ScalingAR’s computation costs, showing that ScalingAR achieves significant performance gains with modest additional computational overhead compared to baselines.
> 5. **Novelty of ScalingAR `Weakness 5`** We emphasize that ScalingAR’s novelty lies in its logit-only, image-specific fusion, which enables actionable mechanisms for AR image generation without intermediate renders or external reward models.
> 6. **Component ablations `Question 2`** We add detailed ablation studies for both the Profile and Policy Levels, showing the distinct and complementary contributions of each component to ScalingAR’s overall performance.
>
> Thank you once again for your thorough and thoughtful review. We fully understand that your time is valuable, and we would greatly appreciate it if you could let us know **whether our rebuttal has adequately addressed your concerns** and **whether it might prompt a reconsideration of the score**.
>
> Warm regards,
>
> Authors

---

### Official Review · Reviewer_uqBd · 2025-11-01

**Soundness:** 2
**Presentation:** 2
**Contribution:** 2
**Rating:** 2
**Confidence:** 4

**Summary:**

This paper proposes ScalingAR, a novel framework that applies Test-Time Scaling (TTS) to autoregressive (AR) image generation models based on the Next-Token Prediction (NTP) paradigm. The proposed method uses the model's intrinsic 'token entropy' as a proxy for the quality of the final result, without relying on external reward models or partial decoding. This confidence score is then used to inform policies that adaptively prune low-quality generation trajectories and dynamically schedule Classifier-Free Guidance (CFG) strength, improving final image quality and efficiency.

**Strengths:**

* The paper is commendable for being the first to explore Test-Time Scaling (TTS) specifically for the Next-Token Prediction (NTP) visual autoregressive paradigm, which is a novel and relevant research direction.
* Some experimental aspects, such as the robustness test using "impossible prompts" are noteworthy.

**Weaknesses:**

1. **Unjustified Rationale for Proposed Metrics**
	* The paper's core motivation could be strengthened. There is insufficient direct evidence for the central claim that "high local token entropy necessarily leads to poor final results". The related concept of "Block Stability" is similarly presented as an assumption; the paper does not currently provide a clear causal link for why a few high-entropy blocks would result in global semantic corruption. These foundational assumptions would be more convincing with direct experimental validation.

2. **Excessive Complexity and Hyperparameter Obscurity**
	* The proposed methodology is quite complex. While the basic concept is understandable, it is questionable whether such intricate formulations and the vast number of associated hyperparameters are necessary.
	* This complexity is a concern because the paper provides limited discussion on the hyperparameter tuning process (e.g., the tuning grid, configurations tested, or associated costs). The sensitivity analysis in Figure 8 suggests performance is notably reliant on this tuning, raising a concern that the reported performance might be more attributable to extensive, model-specific tuning rather than the intrinsic robustness of the method itself.
	* Furthermore, this complexity directly impacts clarity. The paper relies solely on textual descriptions to explain the intricate inference-time process, particularly the interaction between the Profile and Policy levels. The omission of a flow diagram or, at a minimum, a pseudo-code algorithm block makes the method's practical operation difficult to grasp and verify.

3. **Insufficient Experimental Validation and Lack of Transparency**
	* The experimental validation could be more thorough, as several key details are missing, making it difficult to assess the method's contributions fully.
	* **Generalizability**: The method's generalizability is not fully demonstrated. It is only tested on two models (AR-GRPO and LlamaGen). Given the method's apparent dependency on hyperparameters, it would be essential to show that it is portable to other SOTA models (e.g., Emu3, Janus), which were listed but not tested with this method.
	* **Baseline Opacity and Counter-intuitive Results**: The experimental comparison is weakened by a lack of transparency. The paper does not specify the details for the baselines (IS, BoN), such as the number of samples $N$ used, the CFG scales, or the "Best" selection metric. This opacity is concerning, especially given the counterintuitive results in Table 1 (ScalingAR > BoN). Theoretically, a well-implemented BoN (selecting from $N$ full images) should represent a performance upper bound for ScalingAR (pruning from $N$ incomplete images). This discrepancy warrants further explanation.
	* **Insufficient Ablation Study**: The provided ablation (Table 2) is helpful but incomplete, as it only validates the input metrics (Sec 4.1). It does not analyze the policy components (Sec 4.2). A crucial ablation comparing Baseline vs. + Adaptive Termination Gate Only vs. + Guidance Scheduler Only vs. Full ScalingAR is missing. Without this, it is difficult to disentangle the contributions of the two policies and justify the inclusion of the highly complex components.

**Questions:**

1. A key question is whether this framework's philosophy is somewhat at odds with traditional TTS/BoN. Standard TTS/BoN uses computation to explore diversity and then select the best. This method, conversely, appears to prune trajectories that deviate and enforce model certainty. While this is more efficient, the paper should provide clearer intuition for why this pruning strategy should yield a higher-quality ceiling than a full (and costly) BoN exploration.

2. Regarding Figure 8 (Left), the relationship between the pruning threshold $p$ and accuracy is unclear. The chart shows $p=0.15$ (less pruning) resulting in lower accuracy than $p=0.20$ (more pruning). It would be expected that accuracy would monotonically increase as pruning decreases (i.e., $p=0.15$ should be better than $p=0.20$), as the sample set gets closer to the full, unpruned BoN set. Clarification on this relationship is requested.

3. $C_i$ is introduced in Sec.3 but appears unused in the methodology. Additionally, $\hat{D}_t$ is never formally defined. Is this a typo for $D_t$(eq.11)?

---

> ### Author Response · Authors · 2025-11-20
> **[Part 1/3] Response to Reviewer uqBd**
>
> Thank you for your valuable feedback on our manuscript! We have taken your comments seriously and have made the necessary revisions and additions to address the concerns raised. Below is our point-by-point rebuttal:
>
> ---
> > `Weakness 1`: There is insufficient direct evidence for the central claim that "high local token entropy necessarily leads to poor final results".
>
> We sincerely appreciate your thoughtful feedback! We respectfully clarify that our work **does not claim that high entropy necessarily leads to poor final results**. Instead, we treat **high entropy/low confidence as an actionable risk signal** that helps identify high-leverage intervention points during AR generation.
>
> Following your suggestion, we have added a dedicated section with visualization in `Appendix C` to present our key motivation in clear and concise bullet points here:
>
> - **Clarification of Central Claim:** High entropy is not a definitive predictor of poor results but rather an indicator of regions with higher uncertainty and instability. These regions are more likely to contribute to undesirable outcomes, making them valuable targets for intervention.
> - **Supporting Evidence:** Prior studies [1, 2, 3] have shown that high local entropy correlates with increased error probability, particularly in dense information regions prone to instability. For instance, [1] demonstrates that high-entropy tokens are associated with higher error likelihoods in AR tasks; RL studies [2, 3] highlight that high-entropy states often lead to unstable decisions, underscoring the importance of entropy management for stable reasoning and generation. These findings validate our use of entropy as a practical signal for identifying high-risk regions.
> - **ScalingAR's Contribution:** ScalingAR leverages these confidence signals to prune low-confidence trajectories and stabilize generation. By dynamically managing high-entropy regions, it reduces the risks of semantic corruption while maintaining broad applicability across diverse models. Our consistent performance improvements across benchmarks demonstrate the effectiveness of this approach.
> - **Visualization of Token Entropy:** To further address your concern, we have added a dedicated visualization in `Appendix C` (`Figure 11`) to illustrate the relationship between token entropy and generation quality. These visualizations show that regions with poor generation quality often correspond to higher entropy, reinforcing our claim that high-entropy tokens are actionable risk signals. By stabilizing these regions, ScalingAR improves overall output quality.
>
> We hope these clarifications and additional evidence address your concerns. Thank you again for your valuable feedback, which has helped us strengthen the presentation of our work!
>
> ---
> > `Weakness 2`: Excessive Complexity and Hyperparameter Obscurity.
>
> Thank you for your thoughtful feedback! Below, we provide detailed responses:
>
> - **On Methodology Complexity:** While our methodology involves multiple components, its design is motivated by our early exploration, where **we found these parameters to be effective for improving AR image generation**. Our goal is to provide the community with actionable insights for managing uncertainty and instability in AR tasks. The sensitivity analysis in `Figure 8` further validates the importance of these parameters. To address your concern, we have added a more comprehensive hyperparameter sensitivity analysis in `Appendix A.2`, covering token-level confidence/worst-block stability weights, smoothing factors, recovery thresholds, and more.
> - **Hyperparameter Tuning and Robustness:** ScalingAR's hyperparameters are **not model-specific**. In our experiments, the same ScalingAR settings were applied across different base models (*i.e.*, LlamaGen and AR-GRPO). To further demonstrate the robustness of ScalingAR, we extend our evaluation to additional models (*i.e.*, SimpleAR-1.5B [4] and Janus-Pro-1B [5]). As shown in Table A, ScalingAR consistently improves performance across diverse architectures, highlighting its generalizability:
>
>     *Table A: Evaluation of ScalingAR on more base models on GenEval.*
>     ||Two Object|Position|Color Attribute|Overall|
>     |-|-|-|-|-|
>     |SimpleAR [4]|0.90|0.28|0.45|0.63|
>     |**+ ScalingAR (Ours)**|**0.93**|**0.36**|**0.51**|**0.67**|
>     |Janus-Pro [5]|0.82|0.65|0.56|0.73|
>     |**+ ScalingAR (Ours)**|**0.87**|**0.69**|**0.61**|**0.77**|
>
>     This demonstrates that our reported performance gains are not solely attributable to extensive tuning but rather reflect the intrinsic robustness of ScalingAR.
> - **Improving Clarity:** We acknowledge that the textual descriptions of the methodology may be challenging to grasp. To address this, we have added a pseudo-code in `Appendix D` to clearly illustrate the inference-time process, including the interaction between Profile and Policy levels.

---

> ### Author Response · Authors · 2025-11-20
> **[Part 2/3] Response to Reviewer uqBd**
>
> > `Weakness 3.1: Generalizability` The method's generalizability is not fully demonstrated. It is only tested on two models (AR-GRPO and LlamaGen).
>
> To address your concern, we have tested ScalingAR on additional base models (*i.e.*, SimpleAR-1.5B [4] and Janus-Pro-1B [5]) using the same set of parameters, as shown in Table A. These results demonstrate that ScalingAR consistently improves performance across diverse architectures, further validating its generalizability.
>
> ---
> > `Weakness 3.2: Baseline Opacity` The paper does not specify the details for the baselines (IS, BoN).
>
> Thank you for pointing that out, and we apologize for any potential confusion caused! We implement IS and BoN baselines following the settings in TTS-VAR [6], where the reward model used is ImageReward [7], CFG values are set to the default values of the respective base models, and we use $N=8$ as the primary sample number. Thank you again for your valuable feedback!
>
> ---
> > `Weakness 3.3 & Question 1: Counter-intuitive Results with BoN` Theoretically, a well-implemented BoN (selecting from $N$ full images) should represent a performance upper bound for ScalingAR (pruning from incomplete images). This discrepancy warrants further explanation.
>
> We acknowledge your insight. However, we respectfully clarify that ScalingAR's superior performance compared to BoN can be attributed to its design, which **goes beyond simply pruning low-confidence trajectories**. ScalingAR dynamically adjusts the CFG strength through its Guidance Scheduler, enabling more effective exploration and stabilization during generation. In contrast, **BoN relies on fixed trajectories without such dynamic adjustments**.
>
> Additionally, BoN's performance is heavily dependent on the performance of the reward model. To further illustrate this, we present a comparison of BoN using different reward models (*i.e.*, ImageReward, CLIP Score, GPT Score):
>
> *Table B: Performance comparison between TTS strategies with different reward models.*
> ||Reward|Basic|Advanced|Designer|Overall|
> |-|-|-|-|-|-|
> |LlamaGen|-|49.6|40.4|40.3|40.4|
> |+ BoN|CLIP Score|52.0|40.4|36.9|40.8|
> |+ BoN|GPT Score|53.2|41.0|40.7|43.2|
> |+ BoN|ImageReward|54.8|40.8|37.7|42.0|
> |**+ ScalingAR w/o Guidance Scheduler**|-|54.1|43.1|42.2|45.1|
> |**+ ScalingAR (Ours)**|-|**57.4**|**44.1**|**42.5**|**46.5**|
>
> As shown, ScalingAR consistently outperforms BoN across all reward models, with significant gains when the Guidance Scheduler is included. This highlights the importance of ScalingAR's dynamic adjustments and its ability to effectively stabilize generation.
>
> ---
> > `Weakness 3.4: Insufficient Ablation Study` A crucial ablation comparing Baseline vs. + Adaptive Termination Gate Only vs. + Guidance Scheduler Only vs. Full ScalingAR is missing.
>
> Our original intent was to focus on the Profile Level in `Table 2`, as the Policy Level builds upon it (*e.g.*, "w/o Conditional Channel" equals to "Termination Only"). However, to address your concern and further validate the effectiveness of our method, we now provide a dedicated ablation analysis for the Policy Level:
>
> *Table C: Ablation analysis on Policy Level on TIIF-Bench.*
> ||Basic|Advanced|Designer|Overall|
> |-|-|-|-|-|
> |LlamaGen|49.6|40.4|40.3|40.4|
> |+ Termination Only|54.1|43.1|42.2|45.1|
> |+ Scheduler Only|53.6|42.0|41.0|43.8|
> |**+ ScalingAR (Ours)**|**57.4**|**44.1**|**42.5**|**46.5**|
>
> These results demonstrate the individual contributions of the Termination Gate and Guidance Scheduler, as well as their combined effectiveness in ScalingAR. More parameter analyses are provided in `Appendix A.2`.

---

> ### Author Response · Authors · 2025-11-20
> **[Part 3/3] Response to Reviewer uqBd**
>
> > `Question 2`: Regarding Figure 8 (Left), the relationship between the pruning threshold and accuracy is unclear. The chart shows $\mathbf{p}=0.15$ (less pruning) resulting in lower accuracy than $\mathbf{p}=0.20$ (more pruning). It would be expected that accuracy would monotonically increase as pruning decreases (*i.e.*, $\mathbf{p}=0.15$ should be better than ), as the sample set gets closer to the full, unpruned BoN set.
>
> Thank you for your insightful question! We understand that intuitively, one might expect accuracy to monotonically improve as pruning decreases (*i.e.*, lower $\mathbf{p}$ values corresponding to less pruning should yield better accuracy). However, the relationship between $\mathbf{p}$ and accuracy is not monotonic due to the selective nature of our pruning, which is coupled with early termination and CFG scheduling.
>
> Here, $\mathbf{p}$ represents a block-entropy $\mathbf{p}$-quantile (higher $\mathbf{p}$ $\rightarrow$ stricter pruning). The observed behavior can be explained as follows:
>
> - At $\mathbf{p}= 0.15$, **under-pruning** occurs: borderline, low-utilization/unstable trajectories survive, consuming tokens and causing CFG over-correction, which slightly reduces accuracy.
> - At $\mathbf{p}=0.25$, **over-pruning** occurs: diversity shrinks, and some promising but initially noisy paths are prematurely removed.
> - At $\mathbf{p}≈0.20$, a balance is struck: early removal of bad-in-expectation paths reduces interference and reallocates tokens to stronger candidates, yielding the highest accuracy.
>
> This selective pruning mechanism ensures that accuracy peaks at the optimal threshold, rather than monotonically increasing with less pruning.
>
> ---
> > `Question 3`: $C_i$ is introduced in Sec.3 but appears unused in the methodology. Additionally, $\widehat{D}_t$ is never formally defined. Is this a typo for $D_t$ (eq.11)?
>
> Thank you for your detailed review! The $C_i$ introduced in Section 3 is intended to provide additional background context and is not explicitly used in the later methodology. Regarding your second point, you are correct that $D_t$ in Eq. (11) is a typo and should be $\widehat{D}_t$. We have corrected this in the revised manuscript and clarified the definition for consistency. We appreciate your careful attention to detail and thank you for bringing this to our attention!
>
> ---
> [1] On the Entropy Calibration of Language Models, NeurIPS 2025
>
> [2] The Entropy Mechanism of Reinforcement Learning for Reasoning Language Models
>
> [3] Beyond the 80/20 Rule: High-Entropy Minority Tokens Drive Effective Reinforcement Learning for LLM Reasoning, NeurIPS 2025
>
> [4] SimpleAR: Pushing the Frontier of Autoregressive Visual Generation through Pretraining, SFT, and RL
>
> [5] Janus-Pro: Unified Multimodal Understanding and Generation with Data and Model Scaling
>
> [6] TTS-VAR: A Test-Time Scaling Framework for Visual Auto-Regressive Generation, NeurIPS 2025
>
> [7] ImageReward: Learning and Evaluating Human Preferences for Text-to-Image Generation, NeurIPS 2023

---

> ### Author Response · Authors · 2025-11-25
> **Thank you & Looking forward to further discussion**
>
> Dear Reviewer uqBd,
>
> We extend our highest respect for the time and effort you devoted to reviewing our manuscript. We can truly sense the considerable attention you have given to every detail. To facilitate better communication and express our gratitude, we sincerely summarize our rebuttal as follows:
>
> 1. **High local token entropy and poor results `Weakness 1`** We clarify that high entropy is treated as an actionable risk signal, not a definitive predictor of poor results. Supporting evidence, additional visualizations, and references are provided to strengthen this point.
> 2. **Method complexity and hyperparameter tuning `Weakness 2`** We add detailed sensitivity analyses and pseudo-code to improve clarity. ScalingAR's robustness across diverse models is demonstrated with consistent results using the same hyperparameters.
> 3. **Generalizability of ScalingAR `Weakness 3.1`** ScalingAR is tested on additional models (SimpleAR-1.5B and Janus-Pro-1B), showing consistent performance improvements and validating its generalizability.
> 4. **Baseline implementation details `Weakness 3.2`** We clarify the implementation of IS and BoN baselines, following TTS-VAR, and provide detailed configurations for reproducibility.
> 5. **Counter-intuitive BoN results `Weakness 3.3, Question 1`** We explain that ScalingAR's dynamic adjustments (*e.g.*, Guidance Scheduler) enable it to outperform BoN, which relies on fixed trajectories. Additional comparisons with different reward models further validate this.
> 6. **Ablation study on ScalingAR components `Weakness 3.4`** A dedicated ablation study is added, showing the individual and combined contributions of the Termination Gate and Guidance Scheduler.
> 7. **Pruning threshold *vs.* accuracy `Question 2`** We clarify that the relationship between pruning thresholds and accuracy is non-monotonic due to selective pruning and CFG scheduling, with optimal performance achieved at $\mathbf{p}=0.20$.
> 8. **Clarifications on $C_i$ and $\widehat{D}_t$ `Question 3`** We correct the typo for $\widehat{D}_t$ and clarify that $C_i$ is background context not explicitly used in later methodology.
>
> Thank you once again for your dedication throughout the review process. We fully and humbly understand that you might have a tight schedule, and we would be sincerely grateful if you could let us know **whether our rebuttal has sufficiently addressed your concerns** and **whether it might warrant a reconsideration of the score**. Thank you immensely!
>
> Warm regards,
>
> Authors

---

### Official Review · Reviewer_ysTJ · 2025-11-01

**Soundness:** 4
**Presentation:** 4
**Contribution:** 3
**Rating:** 6
**Confidence:** 2

**Summary:**

The authors demonstrate that test-time scaling (TTS) can be made effective for next-token-prediction (NTP) visual autoregressive (AR) models without partial decoding or external rewards. They introduce ScalingAR, which builds a dual-channel confidence profile: an intrinsic channel based on token entropy, margin, and spatial stability, and a conditional channel measuring the marginal impact of textual conditioning. This calibrated confidence guides two policies: (i) early termination of low-confidence trajectories and (ii) dynamic scheduling of classifier-free guidance (CFG). Figures 1 and 3 illustrate confidence trajectories and the quality–alignment trade-off motivating adaptive guidance.

**Strengths:**

* Clear motivation. Language‑model TTS doesn’t transfer to images due to holism, objective ambiguity, and early‑signal scarcity; their design addresses each explicitly.

* Two‑level design (profile/policy) is principled and easy to plug in; the conditional channel provides a concrete handle on prompt adherence during generation.

* Experimental results are quite comprehensive on standard AR image benchmarks, and the figures show consistent improvements over baselines.

**Weaknesses:**

I'm not an expert of this area, but I have some concerns about the paper based on my understanding.

* In terms of novelty, token-entropy/margin as a confidence proxy is known in language models. In this context, what is new beyond adapting them to image domains? Please clarify what is fundamentally novel in the intrinsic/conditional fusion and how it differs from earlier entropy-based routing in decoding.

* Early termination and confidence-guided re-weighting can introduce extra computations. What is the wall-clock overhead vs. sampling, and how much of the reported improvement gains stem from increased width vs. better selection?

* Regarding sensitivity and robustness, the profile mixes entropy, margin with weights (\alpha_H, \alpha_M), and thresholds for termination. How sensitive are results to these hyper‑parameters and to token‑vocabulary size? The ablation studies on these parameters should be included I think.

**Questions:**

Please refer to weaknesses.

---

> ### Author Response · Authors · 2025-11-20
> **[Part 1/3] Response to Reviewer ysTJ**
>
> Thank you for your thorough review and insightful comments on our manuscript! We have carefully considered your feedback and have made the following revisions and clarifications to address the raised concerns.
>
> ---
> > `Weakness 1`: What is new beyond adapting token-entropy/margin as a confidence proxy to image domains?
>
> Thank you for the insightful question. While token entropy and margin as proxies for confidence are well-established in language models, we respectfully emphasize that **our contribution lies in adapting probability-only signals for next-token prediction AR image generation**, where partial prefixes are unusable and external rewards are undesirable. The novelty of our approach is threefold:
>
> - **Intrinsic profiling without intermediate decoding:** We introduce worst-block spatial stability, which tracks top‑$q\%$ highest-entropy blocks with rolling min–max normalization to capture localized instability in the visual token grid—a signal unique to image generation; Temporal robustness is added through EMA, running-min tracking, and rebound scoring ($R_t$), stabilizing confidence signals over time.
> - **Conditional channel beyond entropy:** We measure prompt utilization via KL divergence ($\mathrm{KL}(p_c\parallel p_u)$) between conditional and unconditional logits, calibrated with rolling $\mathbf{z}$-scores and clipping. This disentangles intrinsic stability from conditioning effectiveness, unlike single-channel entropy gates.
> - **From fusion to policies tailored to NTP:** The unified confidence state drives (i) adaptive termination with rebound-based recovery to avoid false positives and (ii) phase-aware CFG scheduling, dynamically adjusting guidance strength based on prompt utilization and volatility signals.
>
> Unlike prior entropy-based decoding/reranking methods in language models or TTS-VAR (which rely on intermediate renders and external rewards), **ScalingAR is the first probability-only test-time scaling framework for NTP AR image generation**, improving quality while significantly reducing token consumption.
>
> ---
> > `Weakness 2.1`: Early termination and confidence-guided re-weighting can introduce extra computations. What is the wall-clock overhead *vs.* sampling?
>
> Thank you for raising this question. Below, we analyze the computational overhead of our ScalingAR in detail:
>
> - **Per-Step Overhead:** ScalingAR operates on logits already computed during sampling, adding lightweight operations such as entropy/top‑$2$ margin calculations. These operations contribute an additional $O(V)\rightarrow O(k)$ complexity per step, which is negligible in practice.
> - **Overall Wall-Clock Time:** While ScalingAR introduces a slight per-step overhead (+$0.004$–$0.005$ seconds per step), early termination effectively reclaims compute by pruning low-confidence trajectories. As shown in Table A, this results in a significant reduction in overall wall-clock time and FLOPs compared to traditional $N=8$ scaling, while achieving better performance.
>
> *Table A: Computation consumption comparison on GenEval.*
> ||$N$|Avg. Per-step WC (s)|Avg. Overall WC (s)|Token Consumption/Img|FLOPs (TFLOPs)|Memory (GB)|Performance|
> |-|-|-|-|-|-|-|-|
> |LlamaGen|1|0.024|24.93|1024|5.60|6.44|0.32|
> |+ BoN|8|0.025|218.44|8192|39.12|48.72|0.15|
> |**+ ScalingAR (Ours)**|8|0.029|69.56|2350|4.23|18.16|0.36|
>
> In summary, ScalingAR achieves a better trade-off between quality and efficiency, with minimal per-step overhead and significant reductions in overall wall-clock time due to early termination.

---

> ### Author Response · Authors · 2025-11-20
> **[Part 2/3] Response to Reviewer ysTJ**
>
> > `Weakness 2.2`: How much of the reported improvement gains stem from increased width *vs.* better selection?
>
> Thank you for this insightful question. To isolate the impact of better selection versus increased width, we conduct three controlled experiments:
>
> 1. **Matched Width and Budget:** Using the same initial $N=8$ and token cap per sample, as shown in `Figure 5 (Middle)`, ScalingAR improves overall performance on TIIF-Bench by $3.4$–$4.5$ points while consuming significantly fewer tokens ($2.3e+06\rightarrow 8.6e+05$). This demonstrates that the gains stem from better selection rather than wider sampling.
> 2. **Matched Realized Tokens:** We tune BoN/IS with $N=3$ ($\sim 8.4e+05$ tokens) to match ScalingAR's total token consumption ($\sim 8.6e+05$ tokens). As shown in Table B, ScalingAR achieves a +$5.2$–$5.5$ point improvement over BoN/IS, further isolating the contribution of selection quality.
>
>     *Table B: Performance comparison with matched realized tokens on TIIF-Bench.*
>     ||$N$|Token Consumption|Basic|Advanced|Designer|Overall|
>     |-|-|-|-|-|-|-|
>     |LlamaGen|-|2.8e+05|49.6|40.4|40.3|40.4|
>     |+ IS|3|8.4e+05|53.4|39.9|38.8|41.0|
>     |+ IS|8|2.3e+06|54.8|40.3|39.9|42.4|
>     |+ BoN|3|8.4e+05|53.0|40.4|37.3|41.3|
>     |+ BoN|8|2.3e+06|54.8|40.8|37.7|42.0|
>     |**+ ScalingAR (Ours)**|8|8.6e+05|**57.4**|**44.1**|**42.5**|**46.5**|
>
> 3. **Width-Controlled Ablation:** We weaken termination by lowering the confidence threshold ($\mathbf{p}=0.20\rightarrow 0.15$) in `Figure 8 (Left)`. Gains shrink from $46.47$ to $44.16$, confirming that termination and selection are the primary drivers of ScalingAR's improvements.
>
> In conclusion, ScalingAR's performance gains primarily stem from **better selection mechanisms, not increased width**, as evidenced by its efficiency and consistent improvements across controlled settings.

---

> ### Author Response · Authors · 2025-11-20
> **[Part 3/3] Response to Reviewer ysTJ**
>
> > `Weakness 3`: How sensitive are results to the profile mixes entropy, margin with weights ($\alpha_H$, $\alpha_M$), thresholds for termination and to token‑vocabulary size?
>
> Thank you for your valuable suggestion! Following your feedback, we conduct further sensitivity analyses and include detailed results in `Appendix A.2`. Below is a summary:
>
> 1. **Sensitivity to Intrinsic Channel Mixing ($\alpha_H$, $\alpha_M$) and Stability Weights ($w_\text{tok}$, $w_\text{blk}$):** Table C shows that varying $\alpha_H:\alpha_M$ from $0.3:0.7$ to $0.7:0.3$ changes the overall performance by $≤0.7$ points, with the best balance near $0.5:0.5$, indicating that both entropy and margin signals are useful within a reasonable range. Similarly, adjusting $w_\text{tok}$:$w_\text{blk}$ from $0.35:0.65$ to $0.85:0.15$ also results in $≤0.7$ fluctuation, as shown in Table D, showing that token-level and worst-block cues are interchangeable to some extent.
>
>     *Table C: Sensitivity of $\alpha_H$/$\alpha_M$ on TIIF-Bench.*
>     |$\alpha_H$/$\alpha_M$|Basic|Advanced|Designer|Overall|
>     |-|-|-|-|-|
>     |0.3/0.7|56.2|42.6|41.4|45.8|
>     |**0.5/0.5 (Ours)**|**57.4**|**44.1**|**42.5**|**46.5**|
>     |0.7/0.3|56.8|43.6|41.8|46.0|
>
>     *Table D: Sensitivity of $w_\text{tok}$/$w_\text{blk}$ on TIIF-Bench.*
>     |$w_\text{tok}$/$w_\text{blk}$|Basic|Advanced|Designer|Overall|
>     |-|-|-|-|-|
>     |0.35/0.65|55.9|43.1|41.4|45.8|
>     |**0.65/0.35 (Ours)**|**57.4**|**44.1**|**42.5**|**46.5**|
>     |0.85/0.15|57.0|43.5|41.8|46.1|
>
> 2. **Sensitivity to Termination Gate Parameters ($\mathbf{p}$-quantile and $\delta_\text{rec}$):** We have already analyzed both parameters in `Figure 8`, and here we provide further details. As shown in Table E, varying the confidence threshold $\mathbf{p}$ from $0.15$ to $0.25$ affects token consumption and performance, with the best results at $\mathbf{p} = 0.20$. Similarly, Table F shows that adjusting the recovery threshold $\delta_\text{rec}$ from $0.03$ to $0.10$ yields fluctuations of $≤0.9$ points, confirming the robustness of our termination mechanism.
>
>     *Table E: Sensitivity of Confidence Threshold $\mathbf{p}$ on TIIF-Bench.*
>     |$\mathbf{p}$|Token Consumption|Basic|Advanced|Designer|Overall|
>     |-|-|-|-|-|-|
>     |0.15|1.3e+06|54.1|42.4|41.8|44.2|
>     |**0.20 (Ours)**|8.6e+0.5|**57.4**|**44.1**|**42.5**|**46.5**|
>     |0.25|6.9e+05|55.5|42.7|41.8|44.6|
>
>     *Table F: Sensitivity of Recovery Threshold $\delta_{rec}$ on TIIF-Bench.*
>     |$\delta_{rec}$|Token Consumption|Basic|Advanced|Designer|Overall|
>     |-|-|-|-|-|-|
>     |w/o $\delta_{rec}$|8.2e+05|56.4|43.3|41.4|45.8|
>     |0.03|8.9e+05|56.1|43.0|41.8|45.6|
>     |**0.05 (Ours)**|8.6e+0.5|**57.4**|**44.1**|**42.5**|**46.5**|
>     |0.10|8.4e+05|56.7|43.3|41.8|46.0|
>
> 3. **Sensitivity to Token-Vocabulary Size:** While our base models (*i.e.*, LlamaGen, AR-GRPO) use fixed codebook settings, we extended experiments to additional models (*i.e.*, SimpleAR-1.5B [1], Janus-Pro-1B [2]) to validate ScalingAR's generalizability. As shown in Table G, ScalingAR consistently improves performance across different vocabulary sizes and model architectures, further demonstrating its robustness.
>
>     *Table G: Evaluation of ScalingAR on more base models on GenEval.*
>     ||Codebook Size|Two Object|Position|Color Attribute|Overall|
>     |-|-|-|-|-|-|
>     |SimpleAR [1]|64000|0.90|0.28|0.45|0.63|
>     |**+ ScalingAR (Ours)**|64000|**0.93**|**0.36**|**0.51**|**0.67**|
>     |Janus-Pro [2]|16384|0.82|0.65|0.56|0.73|
>     |**+ ScalingAR (Ours)**|16384|**0.87**|**0.69**|**0.61**|**0.77**|
>
> In summary, ScalingAR demonstrates robustness across a range of hyperparameters and vocabulary sizes, with detailed ablation studies included in `Appendix A.2`. Further qualitative results are provided in `Appendix B`.
>
> ---
> [1] SimpleAR: Pushing the Frontier of Autoregressive Visual Generation through Pretraining, SFT, and RL
>
> [2] Janus-Pro: Unified Multimodal Understanding and Generation with Data and Model Scaling

---

> ### Author Response · Authors · 2025-11-25
> **Thank you & Looking forward to further discussion**
>
> Dear Reviewer ysTJ,
>
> We would like to extend our heartfelt thanks to you for your time and efforts in engaging with the author-reviewer discussion. To facilitate a better understanding of our rebuttal and revisions, we summarize your key concerns and our corresponding responses as follows:
>
> 1. **What is new beyond adapting token-entropy/margin to image domains? `Weakness 1`** We clarify that ScalingAR introduces novel contributions, including intrinsic profiling without intermediate decoding, a conditional channel beyond entropy, and NTP-specific fusion and policies.
> 2. **What is the computational overhead of ScalingAR? `Weakness 2.1`** We provide detailed analysis showing that ScalingAR introduces negligible per-step overhead while significantly reducing overall wall-clock time through early termination.
> 3. **Do performance gains stem from increased width or better selection? `Weakness 2.2`** Controlled experiments confirms that ScalingAR's improvements primarily stem from better selection mechanisms, not increased width.
> 4. **How sensitive are results to hyperparameters and vocabulary size? `Weakness 3`** Extensive sensitivity analyses show that ScalingAR is robust to hyperparameter variations and consistently improves performance across different vocabulary sizes and models.
>
> For other issues not mentioned here, please refer to our detailed rebuttal response. **We sincerely hope this addresses your concerns!** We respectfully look forward to further discussion with you.
>
> Warm regards,
>
> Authors

---

### Author Response · Authors · 2025-11-20
**Summary of Manuscript Revision**

Dear Reviewers,

We sincerely thank all the reviewers for your thoughtful and constructive feedback! We are really encouraged to see that the reviewers appreciate several positive aspects of our work, including **clear motivation & practical design** (Reviewer `ysTJ`), **novel framework** (Reviewer `uqBd`), **well-scoped claim** (Reviewer `E86T`), **comprehensive & noteworthy experiments** (Reviewer `ysTJ`, `uqBd`), as well as **effective & efficient method addressing key challenges** in AR image generation (Reviewer `uqBd`, `E86T`).

Your expertise has been invaluable in helping us strengthen our manuscript! In addition to addressing your comments point-by-point on the OpenReview forum, we have made the following key revisions to the manuscript (all updated text is highlighted in blue):

- **Expanded Hyperparameter Analysis:** We have included a more comprehensive analysis of ScalingAR's hyperparameters in `Appendix A.2`.
- **Efficiency Evaluation:** Detailed computational cost metrics for ScalingAR have been added in `Table 4`.
- **Transferability Demonstration:** To validate the generalizability of ScalingAR, we have incorporated results on two additional models (*i.e.*, SimpleAR and Janus-Pro) in `Appendix B`.
- **Motivation and Entropy Mechanism:** We have further elaborated on the role of the entropy mechanism in AR image generation, providing additional insights in `Appendix C`.
- **Algorithm Workflow:** We have summarized the workflow of ScalingAR in the form of pseudocode in `Appendix D`.

We have made earnest efforts to address the primary concerns raised and believe these revisions significantly enhance the clarity and comprehensiveness of our manuscript. We respectfully look forward to further feedback to continue improving the quality of our work.

Sincerely,

Authors

---

### Author Response · Authors · 2025-11-23
**General Response**

Dear Reviewers,

Thank you for your thorough and insightful reviews. We sincerely appreciate your feedback, which has been invaluable in improving the quality of our paper. Below, we summarize the key concerns raised and our corresponding responses:

1. **Expanded Hyperparameter Analysis**  (`Reviewers ysTJ, uqBd, E86T`): We have conducted a more detailed sensitivity analysis, including token-level confidence/worst-block stability weights, smoothing factors, recovery window, and warm-up period.
2. **Efficiency Evaluation** (`Reviewers ysTJ, E86T`): Additional details on the efficiency of ScalingAR have been provided, including per-step/overall wall-clock time, FLOPs, and memory usage.
3. **Transferability Demonstration** (`Reviewers ysTJ, uqBd, E86T`): We have demonstrated the robustness and transferability of ScalingAR by evaluating it on two additional models (SimpleAR-1.5B and Janus-Pro-1B) without model-specific hyperparameter tuning, using only the default ScalingAR setting.
4. **Motivation and Entropy Mechanism** (`Reviewer uqBd`): We have elaborated on the role and design of the entropy mechanism in the context of AR image generation.
5. **Algorithm Workflow** (`Reviewer uqBd`): We have summarized the workflow of ScalingAR in the form of pseudocode to enhance clarity.

Thank you again for your valuable feedback. We remain eager to address any further questions or concerns you may have.

Warm regards,

Authors

---

### Meta-Review · Area_Chair_wePW · 2026-01-08

**Summary:**

The paper demonstrates the efficacy of test time scaling (TTS) in next token prediction for visual autoregressive models without partial decoding or external rewards. The proposed method, namely ScalingAR, uses the model's intrinsic 'token entropy' as a proxy for the quality of the final result, without relying on external reward models or partial decoding. The paper received mixed reviews during the initial discussions, and the authors provided a comprehensive rebuttal to address few key concerns raised. However, I believe there are certain points that require more thoughtful addressal. For example,
1.	Significant amount of hyperparameter tuning requirement, though the authors argued these parameters are model agnostic. However, whether the use of same set of hyperparameter can yield near optimal performance, remains an open ended question.
2.	Counter-intuitive Results with BoN, as acknowledged by the authors this may require further investigation.
3.	Concern over selective pruning to not yield monotonic accuracy behavior remains a concern. The authors’ response is not conclusive enough.
Given the method’s heavy reliance on various tunable components and presence of many moving parts, I believe this draft along with the consideration of rebuttal response is still slightly below the acceptance threshold of ICLR.

**Reviewer Concerns:**

The key concerns were the significant hyperparameter dependence of the method and many moving parts, with the accuracy may often requiring hand tuning of various params.

**Reviewer Scores:**

Reviewer ysTJ: 6
Reviewer uqBd: 4
Reviewer E86T: 4

---

### Decision · Program_Chairs · 2026-01-26

Reject